# FROM FORGETTING TO ROBUSTNESS: ROBUST CLASS -INCREMENTAL LEARNING WITH CLIP

## ABSTRACT

Class-Incremental Learning (CIL) aims to enable a model to continuously recognize new categories without forgetting previously learned ones. While most existing methods focus on alleviating catastrophic forgetting, they largely overlook the vulnerability of CIL models to adversarial perturbations, which poses a critical threat to their reliability in real-world applications. Motivated by this oversight, we formalize a new problem setting, Robust Class-Incremental Learning (RCIL). To address the conflict between adversarial robustness and class-incremental learning, we propose **S**elective parameter optimization for **A**dversarial training with **GE**ometric constraint (**SAGE**), which selectively updates critical parameters to protect knowledge learned from previous tasks. Beyond parameter efficiency, SAGE introduces a theoretically grounded geometric constraint together with a contrastive loss to preserve structural relationships among features. This design enables stable and robust learning across tasks under adversarial attacks. Extensive experiments demonstrate that SAGE effectively improves adversarial robustness while mitigating catastrophic forgetting, leading to more reliable and practical CIL models. *The code is provided in the supplementary material.*

## 1 INTRODUCTION

Real-world systems often operate in dynamic environments where new classes appear sequentially and models must maintain performance on previously learned tasks. To address this, Class-Incremental Learning (CIL) has been proposed as a learning paradigm that enables models to adapt to new tasks while retaining knowledge from previous ones, without requiring access to all historical data (Li & Hoiem, 2017; Shin et al., 2017; Wang et al., 2022; Huang et al., 2024). Owing to its effectiveness and practicality, CIL has been widely adopted across diverse domains, including image classification (Kirkpatrick et al., 2017; Yu et al., 2020), image captioning (Nguyen et al., 2019; Del Chiaro et al., 2020), and other vision-language tasks (Greco et al., 2019; Zhang et al., 2023; Wu et al., 2025; Lin et al., 2025). The emergence of large-scale pre-trained vision-language models (Jia et al., 2021; Radford et al., 2021a;b; Yao et al., 2021), most notably CLIP (Radford et al., 2021a), provides a new direction for advancing CIL. Leveraging its strong generalization and rich multimodal representations, CLIP naturally emerged as a promising backbone for CIL (Thengane et al., 2022), significantly outperforming conventional models. Building on this foundation, recent studies (Yu et al., 2024a; Huang et al., 2024; Wang et al., 2023; Jha et al., 2024) further enhance CLIP-based CIL by employing advanced regularization techniques, lightweight adapter modules, or textual representations to mitigate forgetting and improve knowledge transfer.

Despite these advances in alleviating catastrophic forgetting, a critical vulnerability has been largely overlooked, which is susceptibility to adversarial perturbations. These perturbations, crafted deliberately by adversarial attacks, are often imperceptible to humans yet can cause models to produce incorrect predictions, leading to significant accuracy degradation (Szegedy et al., 2014; Madry et al., 2018; Croce & Hein, 2020). Although techniques such as adversarial training (AT) (Zheng et al., 2016; Shafahi et al., 2019b; Zhang et al., 2019) aim to enhance robustness by exposing models to adversarial examples, the objectives of CIL and adversarial robustness are fundamentally distinct. As a result, directly applying AT to CIL can result in suboptimal robustness while significantly exacerbating catastrophic forgetting. Consequently, there is a need for methods that can simultaneously preserve knowledge across tasks while ensuring robustness against adversarial perturbations. **Robust Class-Incremental Learning (RCIL)** addresses this gap by jointly tackling incremental

learning and adversarial robustness, ensuring that models remain accurate and secure across both clean and adversarial inputs throughout continual learning (Bai et al., 2023; Cho et al., 2025).

However, despite its importance, research on RCIL remains limited. A related but distinct line of research, zero-shot adversarial robustness, partially echoes the objectives of RCIL by seeking both robustness enhancement and knowledge preservation. Methods such as TeCoA (Mao et al., 2023), PMG-AFT (Wang et al., 2024b), and TGA-ZSR (Yu et al., 2024b) presuppose a fixed global label space, where robustness is optimized with full label access, limiting their applicability under continually expanding label spaces. These limitations underscore the need for RCIL-specific approaches. Existing attempts in this direction remain scarce. For instance, TABA (Bai et al., 2023) increases sample diversity via mixup, but this does not directly improve robustness, leaving the model vulnerable to stronger attacks. FLAIR (Cho et al., 2025) enhances robustness by constraining distillation with respect to the discrepancy between clean and adversarial outputs in the current task relative to the previous task, implicitly regularizing gradients and Hessians. However, it only captures changes in model outputs without explicitly considering the feature space structure. Consequently, the model may suffer from feature shift, which can undermine the effectiveness of the distillation.

To address the challenge of simultaneously achieving adversarial robustness and knowledge preservation in CIL, we observed that improving robustness often requires substantial parameter updates to adjust decision boundaries, but such updates can destabilize previously learned knowledge, leading to catastrophic forgetting. To reconcile this conflict, we propose a unified framework that combines selective parameter optimization for adversarial training with a geometric constraint. By updating only the most critical parameters, the model rapidly reduces loss and maintains stability while adapting to new tasks. At the same time, the geometric constraint preserves the structural consistency of representations by aligning adversarial and clean embeddings across tasks. Together, these components provide a principled solution for robust class-incremental learning.

Our main contributions can be summarized as follows:

- To the best of our knowledge, this is the first work to formalize the definition of Robust Class-Incremental Learning (RCIL) and theoretically show that adversarial training, which requires extensive parameter updates, conflicts with the stability needed in class-incremental learning.

- We propose a novel method, SAGE, which selectively updates critical parameters identified via gradient-weight products and incorporates a geometric constraint-based contrastive loss to simultaneously enhance robustness and mitigate forgetting.

- Extensive experiments demonstrate that SAGE not only significantly outperforms naive combinations of class-incremental learning and adversarial training across multiple benchmarks, but also achieves superior performance over existing baselines in robust class-incremental learning.

## 2 RELATED WORK

### 2.1 ADVERSARIAL ROBUSTNESS

Deep neural networks (DNNs) are expected to exhibit robustness to minor natural variations in input and maintain consistent predictions. However, extensive studies (Szegedy et al., 2014; Madry et al., 2018; Carlini & Wagner, 2017; Croce & Hein, 2020) have demonstrated that they are highly vulnerable to adversarial perturbations, which can cause incorrect predictions. To enhance adversarial robustness, various defense methods have been proposed. Adversarial training (AT) (Zheng et al., 2016; Wu et al., 2020; Mao et al., 2023) is the most widely studied and empirically validated defense paradigm. It enhances robustness by generating adversarial examples via various attack methods and incorporating them into the training process, thereby exposing the model to challenging perturbations and improving its resilience. Adversarial purification (Nie et al., 2022; Lee & Kim, 2023; Yang et al., 2022) leverages diffusion models to remove adversarial noise from inputs, enabling downstream classifiers to make accurate predictions on the purified samples. Randomized defenses (Ma et al., 2023; Dong & Xu, 2023) introduce stochasticity into the model architecture or inference process to obfuscate gradients and hinder attack effectiveness. More recently, research attention has shifted toward pre-trained vision-language models such as CLIP, which have attracted wide interest for their strong generalization and zero-shot capabilities. Despite these strengths, recent studies (Mao et al., 2023; Schlarmann et al., 2024) reveal that CLIP remains highly vulnerable

to adversarial attacks, motivating a growing body of research (Li et al., 2024; Wang et al., 2024b; Yu et al., 2024b) on improving the robustness of CLIP-based models.

## 2.2 CLASS-INCREMENTAL LEARNING

Class-Incremental Learning (CIL) aims to continuously learn new tasks while retaining knowledge from previous ones, thereby mitigating catastrophic forgetting. Existing approaches are typically categorized into three main groups. Regularization-based methods (Li & Hoiem, 2017; Kirkpatrick et al., 2017; Zenke et al., 2017; Chaudhry et al., 2018a; Liu et al., 2018; Dhar et al., 2019; Rannen et al., 2017) mitigate forgetting by introducing additional constraints during optimization. Replay-based methods (Chaudhry et al., 2018b; Hou et al., 2019; Rebuffi et al., 2017; Buzzega et al., 2020; Boschini et al., 2022) preserve previous knowledge by revisiting samples from earlier tasks while training on new ones. Architecture-based methods (Mallya & Lazebnik, 2018; Mallya et al., 2018; Fernando et al., 2017; Veniat et al., 2020; Ostapenko et al., 2021) dynamically increase model capacity, allocating separate parameters or structures for each new task to minimize interference. Beyond these conventional strategies, the advent of pre-trained models has opened new possibilities for CIL, with vision-language models such as CLIP offering strong multimodal representations. Consequently, several studies (Thengane et al., 2022; Yu et al., 2024a; Huang et al., 2024; Wang et al., 2023; Jha et al., 2024) have explored CIL with CLIP, leveraging its strong visual representation and rich language semantics to mitigate forgetting and enhance adaptability. Existing CIL methods primarily focus on mitigating forgetting, yet recent studies (Bai et al., 2023; Cho et al., 2025) reveal that they remain highly vulnerable to adversarial examples. To address this issue, Robust Class-Incremental Learning (RCIL) jointly tackles CIL and adversarial robustness. However, these approaches rely on simple techniques and fail to exploit the rich representations of pre-trained models, limiting their resilience against adversarial threats. Therefore, we explore RCIL with CLIP, aiming to harness its strong multimodal representations for improved robustness and reduced forgetting.

## 3 PROBLEM DEFINITION

Robust Class-Incremental Learning (RCIL) extends CIL by explicitly incorporating adversarial robustness (Bai et al., 2023; Cho et al., 2025). As illustrated in Figure 1, its objective is to learn a model that correctly classifies both clean and adversarial inputs across sequential tasks.

To formulate the sequential learning setting, following prior work (Thengane et al., 2022; Yu et al., 2024a), we partition the full dataset $\mathcal{D}$ into a sequence of $t$ disjoint tasks, denoted as $\{\mathcal{D}_1, \mathcal{D}_2, \cdots, \mathcal{D}_t\}$. Each task $\mathcal{D}_t = \{\mathcal{X}_t, \mathcal{Y}_t, \mathcal{P}_t\}$ consists of input samples $\mathcal{X}_t$, their labels $\mathcal{Y}_t$, and the set of text prompts $\mathcal{P}_t$. The class sets across different tasks are strictly non-overlapping, where $\mathcal{Y}_i \cap \mathcal{Y}_j = \emptyset$ ($i \neq j$). During training on task $t$, the model $f_{\theta_t}(\cdot)$ is trained solely on data from the current task $\mathcal{D}_t$, without access to data from earlier tasks $\mathcal{D}_{1:(t-1)} = \cup_{i=1}^{t-1} \mathcal{D}_i$. Building on this standard CIL setup, RCIL further augments the training dataset of each task by incorporating dynamically generated adversarial examples. Specifically, adversarial examples $\boldsymbol{x}_t^{adv}$ are generated from the clean examples $\boldsymbol{x}_t \in \mathcal{X}_t$, and we define the adversarial example set as:

$$\mathcal{X}_t^{adv} = \{\boldsymbol{x}_t^{adv} | \boldsymbol{x}_t^{adv} = \boldsymbol{x}_t + \delta_t^{train}, \|\delta_t^{train}\| \leq \epsilon, \boldsymbol{x}_t \in \mathcal{X}_t\} \quad (1)$$

where $\epsilon$ controls the perturbation budget. The adversarial perturbation $\delta_t^{train}$ is obtained by:

$$\arg \max_{\|\delta_t^{train}\| \leq \epsilon} \mathcal{L}_{\text{CE}}(\boldsymbol{x}_t + \delta_t^{train}, \boldsymbol{y}_t, \boldsymbol{p}_t), \quad \boldsymbol{y}_t \in \mathcal{Y}_{1:t}, \quad \boldsymbol{p}_t \in \mathcal{P}_{1:t} \quad (2)$$

where $\mathcal{L}_{\text{CE}}$ represents the cross-entropy loss, $\mathcal{Y}_{1:t} = \cup_{i=1}^t \mathcal{Y}_i$ and $\mathcal{P}_{1:t} = \cup_{i=1}^t \mathcal{P}_i$ denote the cumulative label and prompt spaces up to task $t$, respectively. Finally, the training dataset for task $t$ is given by $\mathcal{D}_t^{train} = \{\mathcal{X}_t, \mathcal{Y}_t, \mathcal{P}_t\} \cup \{\mathcal{X}_t^{adv}, \mathcal{Y}_t, \mathcal{P}_t\}$.

This allows the model to learn from both clean and adversarial examples simultaneously. Accordingly, the overall training objective at task $t$ integrates the adversarial robustness loss $\mathcal{L}_t^{\text{R}}$ with the CIL loss $\mathcal{L}_t^{\text{CIL}}$ to jointly promote robustness and retention:

$$\mathcal{L}_t^{\text{RCIL}} = \mathcal{L}_t^{\text{R}}(\mathcal{X}_t \cup \mathcal{X}_t^{adv}, \mathcal{Y}_t, \mathcal{P}_t) + \mathcal{L}_t^{\text{CIL}}(\mathcal{X}_t \cup \mathcal{X}_t^{adv}, \mathcal{Y}_t, \mathcal{P}_t) \quad (3)$$

After training on task $t$, the model is evaluated on the joint evaluation set $\mathcal{D}_{1:t}^{eval} = \cup_{i=1}^t \mathcal{D}_i^{eval}$, which includes both clean and regenerated adversarial examples. Importantly, adversarial examples

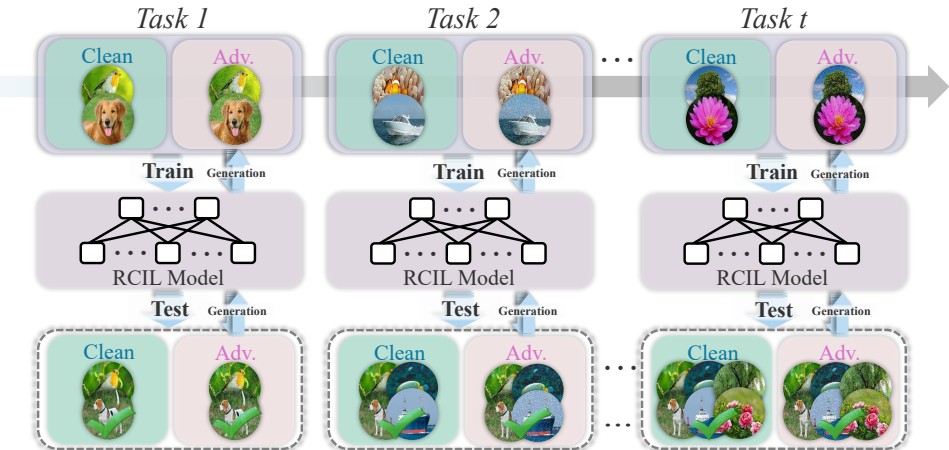

Figure 1: Illustration of Robust Class-Incremental Learning (RCIL). At each task $t$, adversarial examples are generated on the fly, and the model is jointly optimized on both clean and adversarial examples. During evaluation, the test set includes clean examples and regenerated adversarial examples from all classes encountered thus far, reflecting the continually expanding class space.

are recomputed at each evaluation stage under the expanded class space, as shown in Figure 1. A mismatch arises because during training on a previous task $i$ $(i < t)$, adversarial examples are generated using Eq. 1 and 2 with respect to $\mathcal{Y}_{1:i}$. At evaluation stage, however, samples from $\mathcal{D}_i^{eval}$ are perturbed with the enlarged label set $\mathcal{Y}_{1:t}$, still following Eq. 1, but with a new perturbation term:

$$\arg \max_{\|\delta_i^{eval}\| \leq \epsilon} \mathcal{L}_{\mathrm{CE}}(\boldsymbol{x}_i + \delta_i^{eval}, \boldsymbol{y}_i, \boldsymbol{p}_i), \quad \boldsymbol{y}_i \in \mathcal{Y}_{1:t}, \quad \boldsymbol{p}_i \in \mathcal{P}_{1:t} \tag{4}$$

This expansion of the class space significantly increases the challenge for the model, since the regenerated adversarial examples are crafted against the current model and therefore more aggressive.

## 4 METHOD

### 4.1 OBSERVATION

In **class-incremental learning** (Wang et al., 2024a), a key challenge is to preserve performance on previously learned tasks while effectively adapting to new ones. A theoretical motivation for this challenge can be derived by analyzing how parameter updates affect the outputs of old tasks. Specifically, for old-task data $\boldsymbol{x}_{t-1}$, we denote $f_{\theta_{t-1}}(\boldsymbol{x}_{t-1})$ as the model output before the parameter update and $f_{\theta_t}(\boldsymbol{x}_{t-1})$ as the output after updating the parameters from $\theta_{t-1}$ to $\theta_t$. The output difference can be approximated via a first-order Taylor expansion:

$$f_{\theta_t}(\boldsymbol{x}_{t-1}) \approx f_{\theta_{t-1}}(\boldsymbol{x}_{t-1}) + \nabla_{\theta_{t-1}} f_{\theta_{t-1}}(\boldsymbol{x}_{t-1})^\top \Delta\theta \tag{5}$$

where $\Delta\theta = \theta_t - \theta_{t-1}$, and $\nabla_{\theta_{t-1}} f_{\theta_{t-1}}(\boldsymbol{x}_{t-1})$ denotes the gradient of the model output with respect to the parameters at $\theta_{t-1}$. This shows that small parameter updates lead to bounded changes in the outputs on data from previous tasks, helping preserve prior knowledge and mitigate forgetting.

Building on the above discussion, we next consider a complementary perspective from **adversarial robustness**. Its key objective is to reduce the model's sensitivity to input perturbations, thereby ensuring stable predictions and improved robustness (Wu et al., 2020). Formally, given an adversarial input $\boldsymbol{x}_t^{adv} = \boldsymbol{x}_t + \delta$, the model output can be approximated by a first-order Taylor expansion:

$$f_{\theta_t}(\boldsymbol{x}_t + \delta) \approx f_{\theta_t}(\boldsymbol{x}_t) + \nabla_{\boldsymbol{x}_t} f_{\theta_t}(\boldsymbol{x}_t)^\top \delta \tag{6}$$

where $\nabla_{\boldsymbol{x}_t} f_{\theta_t}(\boldsymbol{x}_t)$ represents the gradient of the model output with respect to the input. Since $\delta$ is chosen by the adversary and cannot be directly controlled during training, the key objective becomes reducing the first-order term $\nabla_{\boldsymbol{x}_t} f_{\theta_t}(\boldsymbol{x}_t)$, which measures the model's local sensitivity to input perturbations. Lowering this gradient effectively flattens the loss landscape around the input,

leading to more robust predictions under adversarial noise. In neural networks, the input gradient $\nabla_{\boldsymbol{x}_t} f_{\theta_t}(\boldsymbol{x}_t)$ can be computed through the chain rule across all layers:

$$\nabla_{\boldsymbol{x}_t} f_{\theta_t}(\boldsymbol{x}_t) = \frac{\partial f_{\theta_t}(\boldsymbol{x}_t)}{\partial \boldsymbol{x}_t} = \frac{\partial f_{\theta_t}(\boldsymbol{x}_t)}{\partial \boldsymbol{z}_n} \cdot \frac{\partial \boldsymbol{z}_n}{\partial \boldsymbol{z}_{n-1}} \cdot \ldots \cdot \frac{\partial \boldsymbol{z}_1}{\partial \boldsymbol{x}_t} \tag{7}$$

where $\boldsymbol{z}_n$ denotes the intermediate representation at the $n$-th layer. This formulation reveals that the input gradient depends on the entire forward-backward propagation path through the network. Therefore, effectively reducing $\nabla_{\boldsymbol{x}_t} f_{\theta_t}(\boldsymbol{x}_t)$ often necessitates coordinated updates across multiple layers. In practice, this typically requires extensive parameter updates throughout the network, which is why many adversarial defense methods (Mao et al., 2023; Wang et al., 2024b; Yu et al., 2024b) adopt full-parameter training to modify the network's sensitivity to input perturbations.

Thus, these objectives push feature representations in conflicting directions, leading to a trade-off. This theoretical observation is further validated empirically in Appendix B, where naive combinations of adversarial training and standard CIL significantly increase forgetting or reduce robustness. This conflict reflects both the opposing nature of the optimization objectives and the empirical observations, motivating the need for methods in RCIL that can effectively balance these objectives.

## 4.2 SELECTIVE PARAMETER OPTIMIZATION FOR ADVERSARIAL TRAINING

To balance the competing demands of CIL and adversarial robustness, we adopt a selective parameter optimization strategy. Specifically, training updates only the critical parameters within each layer, thereby balancing two objectives. In this process, important parameters are updated to rapidly reduce loss and enhance robustness, while limiting the scope of updates alleviates feature drift and mitigates forgetting. To identify critical parameters, we quantify their importance using a first-order Taylor approximation of the loss. For a small perturbation $\Delta\theta$, the loss can be approximated as:

$$\mathcal{L}(\theta + \Delta\theta) \approx \mathcal{L}(\theta) + \sum_l \left( \frac{\partial \mathcal{L}(\theta)}{\partial \theta^l} \cdot \Delta\theta^l \right) \tag{8}$$

This formulation allows us to approximate the effect of parameter removal. Following network pruning methods (Sanh et al., 2020), we represent the hypothetical removal of a parameter $\theta^l$ as the perturbation $\Delta\theta^l = -\theta^l$. Substituting this into the above expansion yields the approximate change:

$$\Delta\mathcal{L}_l \approx -\theta^l \cdot \frac{\partial \mathcal{L}(\theta)}{\partial \theta^l} \tag{9}$$

Based on this result, we define the importance of each parameter as the absolute magnitude of the estimated loss change, that is, $I_l = \left| \theta^l \cdot \frac{\partial \mathcal{L}(\theta)}{\partial \theta^l} \right|$. This metric provides a theoretically grounded measure of parameter importance by quantifying how $\theta^l$ affects the loss. In addition, it is simple and practical to compute since it only requires the parameter values and their gradients and does not introduce extra hyperparameters. By selecting and updating only the top-$k\%$ parameters with the highest importance values $I_l$, we strike a balance between two conflicting objectives. To implement this selective update mechanism, we introduce a binary mask $m_l$ defined as:

$$m_l = \begin{cases} 1, & \text{if } I_l \in \text{Top-}k\%(I) \\ 0, & \text{otherwise} \end{cases} \tag{10}$$

Parameter updates are then performed in a masked manner $\theta \leftarrow \theta + m \odot \Delta\theta$, where $\odot$ denotes element-wise multiplication. This approach effectively restricts parameter modifications to those most relevant for the current task, which is crucial for maintaining stability and performance in the class-incremental setting.

## 4.3 GEOMETRIC-CONSTRAINT GUIDED CONTRASTIVE LEARNING

Merely constraining parameter updates is insufficient, since adversarial perturbations can still distort feature representations and compromise consistency across tasks. Prior work in CIL (Kirkpatrick et al., 2017; Zenke et al., 2017) similarly shows that parameter-level regularization only partially prevents parameter drift, but it does not directly address adversarial threats. In particular, without mechanisms to realign perturbed features with their correct class regions, the relational structure

among features may still be severely compromised. This motivates the need to preserve structural relationships within the feature space. To this end, we adopt cosine similarity as a measure of feature alignment, which provides a principled foundation for our loss design.

We first introduce the notation for cosine similarity. For any three unit vectors $\boldsymbol{a}$, $\boldsymbol{b}$ and $\boldsymbol{c}$, we obtain:

$$\gamma_{\boldsymbol{ab}} = \cos(\boldsymbol{a}, \boldsymbol{b}) = \boldsymbol{a}^\top \boldsymbol{b}, \quad \gamma_{\boldsymbol{ac}} = \cos(\boldsymbol{a}, \boldsymbol{c}) = \boldsymbol{a}^\top \boldsymbol{c}, \quad \gamma_{\boldsymbol{bc}} = \cos(\boldsymbol{b}, \boldsymbol{c}) = \boldsymbol{b}^\top \boldsymbol{c} \tag{11}$$

These cosine similarities are not independent but follow a geometric constraint, the proof of which is provided in Appendix C:

$$\gamma_{\boldsymbol{ac}}\gamma_{\boldsymbol{bc}} - \sqrt{(1 - \gamma_{\boldsymbol{ac}}^2)(1 - \gamma_{\boldsymbol{bc}}^2)} \leq \gamma_{\boldsymbol{ab}} \leq \gamma_{\boldsymbol{ac}}\gamma_{\boldsymbol{bc}} + \sqrt{(1 - \gamma_{\boldsymbol{ac}}^2)(1 - \gamma_{\boldsymbol{bc}}^2)} \tag{12}$$

The relations indicate that $\gamma_{\boldsymbol{ab}}$ is bounded by a range determined by $\gamma_{\boldsymbol{ac}}$ and $\gamma_{\boldsymbol{bc}}$. When the constraint is not saturated, $\gamma_{\boldsymbol{ab}}$ admits direct optimization with gradient $\frac{\partial \gamma_{\boldsymbol{ab}}}{\partial \theta}$. However, when the constraint becomes active, $\gamma_{\boldsymbol{ab}}$ is dictated by $\gamma_{\boldsymbol{ac}}$ and $\gamma_{\boldsymbol{bc}}$, and the gradient can be expressed:

$$\frac{\partial \gamma_{\boldsymbol{ab}}}{\partial \theta} = \frac{\partial \mathcal{F}(\gamma_{\boldsymbol{ac}}, \gamma_{\boldsymbol{bc}})}{\partial \gamma_{\boldsymbol{ac}}} \cdot \frac{\partial \gamma_{\boldsymbol{ac}}}{\partial \theta} + \frac{\partial \mathcal{F}(\gamma_{\boldsymbol{ac}}, \gamma_{\boldsymbol{bc}})}{\partial \gamma_{\boldsymbol{bc}}} \cdot \frac{\partial \gamma_{\boldsymbol{bc}}}{\partial \theta} \tag{13}$$

where $\mathcal{F}(\gamma_{\boldsymbol{ac}}, \gamma_{\boldsymbol{bc}})$ denotes the bound $\gamma_{\boldsymbol{ac}}\gamma_{\boldsymbol{bc}} \pm \sqrt{(1 - \gamma_{\boldsymbol{ac}}^2)(1 - \gamma_{\boldsymbol{bc}}^2)}$. Therefore, optimizing $\gamma_{\boldsymbol{ab}}$ implicitly affects $\gamma_{\boldsymbol{ac}}$ and $\gamma_{\boldsymbol{bc}}$ whenever the constraint becomes active.

Specifically, we consider three feature representations: the current model's embedding of the adversarial example $f_{\theta_t}(\boldsymbol{x}_t^{adv})$, the previous model's embedding of the clean examples $f_{\theta_{t-1}}(\boldsymbol{x}_t)$, and the current model's embedding of the clean example $f_{\theta_t}(\boldsymbol{x}_t)$. For clarity, we normalize them as:

$$\boldsymbol{a} = \frac{f_{\theta_t}(\boldsymbol{x}_t^{adv})}{\left\| f_{\theta_t}(\boldsymbol{x}_t^{adv}) \right\|}, \quad \boldsymbol{b} = \frac{f_{\theta_{t-1}}(\boldsymbol{x}_t)}{\left\| f_{\theta_{t-1}}(\boldsymbol{x}_t) \right\|}, \quad \boldsymbol{c} = \frac{f_{\theta_t}(\boldsymbol{x}_t)}{\left\| f_{\theta_t}(\boldsymbol{x}_t) \right\|} \tag{14}$$

Here, $\gamma_{\boldsymbol{ac}}$ corresponds to improving robustness, while $\gamma_{\boldsymbol{bc}}$ reflects preserving previously learned knowledge. According to the geometric constraint in Eq. 12, optimizing the similarity $\gamma_{\boldsymbol{ab}}$ implicitly influences both $\gamma_{\boldsymbol{ac}}$ and $\gamma_{\boldsymbol{bc}}$. Therefore, by focusing on optimizing $\gamma_{\boldsymbol{ab}}$, we can simultaneously improve robustness and mitigate forgetting, without the need to explicitly balance $\gamma_{\boldsymbol{ac}}$ and $\gamma_{\boldsymbol{bc}}$.

To fully exploit the relational structure of the embeddings and optimize cosine similarity, we introduce a symmetric contrastive loss. By enforcing this symmetry, the structural consistency of the feature space is strengthened, which is crucial for maintaining cross-task stability while improving adversarial robustness. Formally, given a batch of clean and adversarial examples, the symmetric contrastive loss is defined as:

$$\mathcal{L}_{\text{con}} = \frac{1}{2N} \sum_{i=1}^{N} \left[ -\log \frac{\exp(\boldsymbol{a}_i^\top \boldsymbol{b}_i / \tau)}{\sum_{j=1}^{N} \exp(\boldsymbol{a}_i^\top \boldsymbol{b}_j / \tau)} - \log \frac{\exp(\boldsymbol{b}_i^\top \boldsymbol{a}_i / \tau)}{\sum_{j=1}^{N} \exp(\boldsymbol{b}_i^\top \boldsymbol{a}_j / \tau)} \right] \tag{15}$$

where $\tau$ is a temperature parameter controlling the sharpness of the similarity distribution.

Finally, the final loss function can be expressed as:

$$\mathcal{L}_{\text{total}} = \mathcal{L}_{\text{CE}}(\boldsymbol{x}^{adv}, \boldsymbol{y}, \boldsymbol{p}) + \mu \cdot \mathcal{L}_{\text{con}} \tag{16}$$

where $\mu$ is a hyperparameter that balances the two loss terms. In addition, the details of the proposed algorithm are provided in Appendix D.

## 5 EXPERIMENTS

### 5.1 EXPERIMENTAL SETUP

**Datasets.** We conduct experiments on CIFAR-10 (Krizhevsky et al., 2009), CIFAR-100 (Krizhevsky et al., 2009), STL-10 (Coates et al., 2011), and Tiny-ImageNet (Le & Yang, 2015). Specifically, CIFAR-10 and STL-10 are split into 5 tasks with 2 classes per task, referred to as S-CIFAR10 and S-STL10. CIFAR-100 is divided into 10 tasks with 10 classes per task, denoted as S-CIFAR100. Tiny-ImageNet is divided into 10 tasks with 20 classes per task, denoted as S-TinyImageNet.

Table 1: Evaluation of several methods on ViT-B/32 without memory. We report Clean, PGD-10, Auto. accuracy (%), and $BWT$ on S-CIFAR10 and S-STL10 under attack strength of 1/255. **Bold** for the best result, underline for secondary.

| Type | Method | S-CIFAR10 | | | | | | | S-STL10 | | | | | | |
|---|---|---|---|---|---|---|---|---|---|---|---|---|---|---|---|
| | | Clean | | | PGD | | | Auto. | Clean | | | PGD | | | Auto. |
| | | $\overline{A}\uparrow$ | $A_{last}\uparrow$ | $BWT\uparrow$ | $\overline{A}\uparrow$ | $A_{last}\uparrow$ | $BWT\uparrow$ | $A_{last}\uparrow$ | $\overline{A}\uparrow$ | $A_{last}\uparrow$ | $BWT\uparrow$ | $\overline{A}\uparrow$ | $A_{last}\uparrow$ | $BWT\uparrow$ | $A_{last}\uparrow$ |
| AT | TeCoA | 35.70 | 10.12 | -69.68 | 18.06 | 9.20 | -41.70 | 0.09 | 93.59 | 86.79 | -15.94 | 40.06 | 30.14 | -34.25 | 0.27 |
| | FARE | **88.69** | **81.75** | **-17.66** | 33.10 | 27.33 | **-15.31** | 2.33 | **96.59** | **94.28** | **-6.52** | 51.57 | 43.02 | -21.42 | 3.33 |
| | PMG-AFT | 35.48 | 10.86 | -68.78 | 17.98 | 9.11 | -41.66 | 0.05 | 93.61 | 86.88 | -15.83 | 39.88 | 30.06 | -34.42 | 0.24 |
| | TGA-ZSR | 52.98 | 28.34 | -84.91 | 30.45 | 15.70 | -65.55 | 10.78 | 90.92 | 81.22 | -22.61 | 50.69 | 41.51 | -41.02 | 16.44 |
| R-CIL | R-LwF | 45.19 | 19.82 | -98.84 | 43.61 | 19.38 | -95.05 | 19.34 | 51.94 | 36.72 | -77.25 | 47.30 | 28.16 | -80.72 | 27.02 |
| | R-LwF-MC | 55.48 | 33.81 | -80.35 | 47.80 | 24.08 | -97.60 | 23.42 | 71.93 | 69.04 | -36.27 | 57.35 | 47.01 | -52.61 | 36.41 |
| | R-EWC-on | 45.00 | 19.74 | -98.35 | 43.47 | 19.38 | -94.40 | 19.38 | 50.08 | 32.15 | -82.25 | 45.48 | 25.59 | -83.45 | 25.30 |
| | R-SI | 45.22 | 19.81 | -98.94 | 43.59 | 19.38 | -94.98 | 19.36 | 54.08 | 36.80 | -77.17 | 48.61 | 27.50 | -81.81 | 26.51 |
| R-CIL-CLIP | R-RAPF | 46.36 | 19.88 | -98.64 | 44.21 | 19.34 | -94.95 | 19.32 | 55.09 | 40.02 | -70.36 | 32.43 | 14.25 | -66.05 | 13.85 |
| | R-SG | 43.07 | 18.22 | -60.66 | 42.99 | 18.46 | -74.25 | 0.00 | 55.09 | 40.02 | -32.98 | 40.19 | 30.19 | -34.80 | 9.71 |
| | R-Proof | 36.11 | 14.15 | -75.04 | 34.01 | 13.61 | -70.79 | 12.93 | 44.46 | 16.44 | -96.16 | 41.73 | 15.29 | -89.88 | 15.25 |
| RCIL | FLAIR | 61.27 | 45.83 | -66.05 | 51.73 | 32.31 | -77.91 | 30.90 | 71.32 | 64.92 | -42.17 | 57.69 | 48.26 | -51.59 | 41.65 |
| RCIL4CLIP | SAGE (ours) | 72.36 ± 0.65 | 63.67 ± 1.60 | -34.12 ± 1.50 | 61.75 ± 0.43 | 47.85 ± 1.15 | -42.27 ± 1.52 | 41.60 ± 0.93 | 73.52 ± 0.49 | 69.56 ±1.09 | -33.36 ± 1.58 | 63.96 ± 0.41 | 54.31 ± 0.40 | -40.20 ± 1.27 | 46.32 ± 0.73 |

Table 2: Evaluation of several methods on ViT-B/32 without memory. We report Clean, PGD-10, Auto. accuracy (%), and $BWT$ on S-CIFAR100 and S-TinyImageNet under attack strength of 1/255. **Bold** for the best result, underline for secondary.

| Type | Method | S-CIFAR100 | | | | | | | S-TinyImageNet | | | | | | |
|---|---|---|---|---|---|---|---|---|---|---|---|---|---|---|---|
| | | Clean | | | PGD | | | Auto. | Clean | | | PGD | | | Auto. |
| | | $\overline{A}\uparrow$ | $A_{last}\uparrow$ | $BWT\uparrow$ | $\overline{A}\uparrow$ | $A_{last}\uparrow$ | $BWT\uparrow$ | $A_{last}\uparrow$ | $\overline{A}\uparrow$ | $A_{last}\uparrow$ | $BWT\uparrow$ | $\overline{A}\uparrow$ | $A_{last}\uparrow$ | $BWT\uparrow$ | $A_{last}\uparrow$ |
| AT | TeCoA | 21.75 | 9.40 | -77.17 | 11.36 | 7.12 | -46.41 | 6.10 | 19.01 | 7.34 | -60.48 | 11.39 | 4.85 | -39.94 | 4.06 |
| | FARE | **65.63** | 48.54 | -16.49 | 23.20 | 16.03 | -8.94 | 4.40 | **59.42** | 49.52 | -14.84 | 23.29 | 20.85 | -7.52 | 10.07 |
| | PMG-AFT | 24.98 | 9.34 | -77.66 | 11.37 | 7.09 | -46.61 | 6.41 | 19.34 | 7.45 | -60.76 | 11.58 | 4.88 | -40.51 | 4.11 |
| | TGA-ZSR | 41.52 | 16.20 | -67.68 | 17.74 | 7.73 | -53.00 | 5.20 | 30.10 | 41.61 | -55.49 | 12.73 | 4.88 | -40.06 | 2.96 |
| R-CIL | R-LwF | 27.74 | 9.51 | -90.83 | 24.41 | 8.72 | -80.98 | 8.70 | 22.33 | 7.69 | -73.96 | 17.27 | 6.17 | -59.33 | 6.17 |
| | R-LwF-MC | 3.07 | 1.00 | **-0.00** | 2.96 | 1.00 | **-0.00** | 1.00 | 1.46 | 0.50 | **-0.00** | 1.46 | 0.50 | **-0.00** | 0.50 |
| | R-EWC-on | 26.71 | 9.27 | -88.89 | 23.43 | 8.50 | -78.47 | 8.50 | 22.36 | 7.87 | -74.26 | 17.30 | 6.10 | -59.37 | 6.09 |
| | R-SI | 28.12 | 9.81 | -90.48 | 24.58 | 8.80 | -81.04 | 8.80 | 22.42 | 7.52 | -73.04 | 17.58 | 6.77 | -59.69 | 8.64 |
| R-CIL-CLIP | R-RAPF | 36.01 | 13.87 | -88.17 | 28.57 | 10.67 | -81.39 | 10.57 | 32.08 | 17.30 | -73.06 | 22.87 | 11.00 | -64.24 | 10.77 |
| | R-SG | 46.36 | 27.52 | -41.42 | 34.37 | 19.00 | -35.81 | 14.44 | 29.53 | 12.88 | -41.48 | 20.01 | 8.83 | -29.77 | 7.06 |
| | R-Proof | 13.88 | 3.59 | -37.31 | 11.67 | 3.72 | -32.87 | 3.39 | 6.33 | 1.93 | -16.66 | 5.47 | 1.86 | -15.04 | 1.56 |
| RCIL | FLAIR | 3.05 | 1.00 | **-0.00** | 2.94 | 1.00 | **-0.00** | 1.00 | 1.46 | 0.50 | **-0.00** | 1.46 | 0.50 | **-0.00** | 0.50 |
| RCIL4CLIP | SAGE (ours) | 63.20 ± 0.54 | 49.02 ± 0.35 | -22.62 ± 0.36 | 48.49 ± 0.37 | 35.59 ± 0.37 | -19.89 ± 0.42 | 28.98 ± 0.41 | 56.14 ± 0.04 | 44.72 ± 0.60 | -13.54 ± 0.89 | 40.21 ± 0.22 | 31.95 ± 0.52 | -9.49 ± 0.56 | 26.18 ± 0.38 |

**Baseline.** We conduct experiments on five types of baselines: AT, R-CIL, R-CIL-CLIP, RCIL, and RCIL4CLIP. AT represents zero-shot adversarial robustness. R-CIL refers to conventional CIL methods enhanced with AT. R-CIL-CLIP denotes CIL methods built upon a CLIP backbone, likewise enhanced with AT. RCIL encompasses robust class-incremental learning approaches. Finally, RCIL4CLIP denotes RCIL methods with CLIP, with our proposed approach as the primary representative. In addition, the training details of all baseline methods are provided in Appendix J.

**Training Details.** We conduct all experiments on a single NVIDIA RTX 3090 GPU. During adversarial training, we utilize $l_\infty$-norm PGD (Madry et al., 2018) with 2 iterations to generate adversarial examples, setting both the attack strength and step size to 1/255. The SGD optimizer is employed to minimize the loss, and the text prompt is set to "This is a photo of {}". The hyperparameters are set to $\mu = 1.0$ and $k = 0.01$, with a learning rate of 0.1, a weight decay of 1e-5, and a batch size of 64. The model is trained for 20 epochs on S-CIFAR10 and S-STL10, and for 50 epochs on S-CIFAR100 and S-TinyImageNet. To evaluate adversarial robustness, we apply $l_\infty$-norm PGD with 10 iterations, using an attack strength and attack step size of 1/255, and AutoAttack (Auto.) (Croce & Hein, 2020) with an adversarial strength of 1/255.

**Evaluation Metric.** Following previous work (Wang et al., 2022; Cho et al., 2025), we evaluate the average incremental accuracy $\overline{A} = \frac{1}{T}\sum_{t=1}^{T} A_t$, where $A_t$ denotes the model's average accuracy across all seen tasks after completing training on the $t$-th task. In addition, we report $A_{last}$, which represents the average accuracy of the model after completing training on the final task. To evaluate forgetting, we adopt the backward transfer (BWT) metric, defined as $BWT = \frac{1}{T-1}\sum_{t=1}^{T-1}(A_{T,t} - A_{t,t})$, where $A_{i,j}$ denotes the test accuracy on task $j$ after completing training on task $i$.

Table 3: Evaluation of several methods on ViT-B/32 without memory. We report average PGD-10, Auto. accuracy (%) and $BWT$ on S-CIFAR10, S-STL10, S-CIFAR100 and S-TinyImageNet under attack strengths of 1/255, 2/255, and 4/255. **Bold** for the best result, underline for secondary.

| Type | Method | S-CIFAR10 | | | | S-STL10 | | | | S-CIFAR100 | | | | S-TinyImageNet | | | |
|---|---|---|---|---|---|---|---|---|---|---|---|---|---|---|---|---|---|
| | | PGD | | | Auto. | PGD | | | Auto. | PGD | | | Auto. | PGD | | | Auto. |
| | | $\overline{A}\uparrow$ | $A_{last}\uparrow$ | $BWT\uparrow$ | $A_{last}\uparrow$ | $\overline{A}\uparrow$ | $A_{last}\uparrow$ | $BWT\uparrow$ | $A_{last}\uparrow$ | $\overline{A}\uparrow$ | $A_{last}\uparrow$ | $BWT\uparrow$ | $A_{last}\uparrow$ | $\overline{A}\uparrow$ | $A_{last}\uparrow$ | $BWT\uparrow$ | $A_{last}\uparrow$ |
| AT | TeCoA | 10.91 | 4.39 | -30.50 | 0.03 | 19.39 | 16.23 | -17.72 | 0.09 | 6.37 | 4.86 | -26.90 | 3.20 | 6.36 | 2.91 | -23.29 | 2.04 |
| | FARE | 13.40 | 10.04 | **-6.90** | 0.78 | 26.09 | 20.24 | **-13.01** | 1.11 | 10.19 | 6.95 | -3.79 | 1.62 | 9.75 | 9.06 | -3.10 | 3.69 |
| | PMG-AFT | 10.56 | 4.61 | -28.99 | 0.02 | 19.21 | 16.10 | -17.44 | 0.08 | 6.41 | 4.83 | -27.19 | 3.32 | 6.43 | 2.94 | -23.41 | 2.05 |
| | TGA-ZSR | 17.46 | 10.04 | -37.73 | 3.77 | 26.90 | 22.15 | -24.83 | 5.53 | 9.13 | 4.74 | -28.68 | 2.49 | 5.93 | 2.72 | -20.32 | 1.34 |
| R-CIL | R-LwF | 42.09 | 19.15 | -91.04 | 14.37 | 43.60 | 22.37 | -82.52 | 15.03 | 22.15 | 8.20 | -73.70 | 6.86 | 13.79 | 5.01 | -48.20 | 4.22 |
| | R-LwF-MC | 44.08 | 20.57 | -89.58 | 15.13 | 46.78 | 31.26 | -51.19 | 15.43 | 2.93 | 1.00 | **-0.00** | 0.97 | 1.46 | 0.50 | **-0.00** | 0.50 |
| | R-EWC-on | 41.63 | 18.96 | -89.73 | 15.24 | 41.67 | 20.67 | -82.83 | 15.56 | 20.48 | 7.72 | -68.31 | 6.76 | 13.72 | 4.82 | -47.85 | 4.16 |
| | R-SI | 42.31 | 19.05 | -91.59 | 14.20 | 44.28 | 21.85 | -83.14 | 14.96 | 22.20 | 8.12 | -73.79 | 6.81 | 13.99 | 5.28 | -48.53 | 5.21 |
| R-CIL-CLIP | R-RAPF | 40.02 | 18.46 | -84.65 | 14.57 | 27.71 | 12.42 | -58.46 | 11.45 | 22.82 | 8.89 | -69.47 | 7.37 | 19.55 | 7.46 | -52.34 | 6.47 |
| | R-SG | 36.57 | 14.42 | -61.68 | 0.00 | 40.65 | 22.90 | -34.55 | 3.39 | 22.71 | 12.18 | -26.77 | 7.02 | 12.56 | 5.51 | -56.63 | 3.79 |
| | R-Proof | 28.63 | 12.77 | -61.38 | 10.87 | 35.21 | 12.43 | -71.66 | 11.83 | 7.89 | 3.18 | -25.23 | 2.33 | 4.30 | 1.68 | -12.69 | 1.10 |
| RCIL | FLAIR | 45.76 | 24.84 | -78.63 | 18.43 | 46.35 | 30.79 | -51.81 | 17.90 | 2.93 | 1.00 | **-0.00** | 0.90 | 1.46 | 0.50 | **-0.00** | 0.50 |
| RCIL4CLIP | SAGE (ours) | **47.28** | **30.24** | -38.95 | **21.30** | **49.97** | **33.91** | -42.18 | **23.37** | **32.54** | **21.59** | -14.36 | **15.08** | **24.71** | **18.83** | -6.58 | **13.37** |

Table 4: Effect on each module. We report the Clean, PGD-10, and Auto. $A_{last}$ on S-CIFAR10 and S-CIFAR100 after fine-tuning with PGD-2. **Bold** for the best result.

(a) Effect of each contrastive loss.

| Contrastive Loss | | | S-CIFAR10 | | | S-CIFAR100 | | |
|---|---|---|---|---|---|---|---|---|
| $L_{con}(\boldsymbol{a},\boldsymbol{b})$ | $L_{con}(\boldsymbol{a},\boldsymbol{c})$ | $L_{con}(\boldsymbol{b},\boldsymbol{c})$ | Clean | PGD | Auto. | Clean | PGD | Auto. |
| | ✓ | | 39.93 | 21.48 | 18.33 | 12.95 | 10.58 | 10.41 |
| | | ✓ | 28.73 | 19.15 | 0.31 | 13.64 | 9.19 | 0.02 |
| | ✓ | ✓ | 62.37 | 20.25 | 5.63 | 43.10 | 20.48 | 11.52 |
| ✓ | | | **63.67** | **47.85** | **41.60** | **49.02** | **35.59** | **28.98** |

(b) Effect of selective parameter optimization strategy.

| Parameter Importance | | | S-CIFAR10 | | | S-CIFAR100 | | |
|---|---|---|---|---|---|---|---|---|
| $\left\|\theta^l \cdot \frac{\partial \mathcal{L}(\theta)}{\partial \theta^l}\right\|$ | $\|\theta^l\|$ | $\left\|\frac{\partial \mathcal{L}(\theta)}{\partial \theta^l}\right\|$ | Clean | PGD | Auto. | Clean | PGD | Auto. |
| | | ✓ | 53.68 | 40.94 | 35.66 | 46.55 | 35.10 | **29.22** |
| | ✓ | | 59.48 | 44.60 | 38.65 | 46.30 | 33.67 | 27.52 |
| | | | 60.77 | 38.73 | 30.14 | 41.19 | 25.38 | 17.84 |
| ✓ | | | **63.67** | **47.85** | **41.60** | **49.02** | **35.59** | 28.98 |

## 5.2 MAIN RESULTS

We conduct experiments on four datasets and report $\overline{A}$, $A_{last}$, and $BWT$ metric for both clean and adversarial examples generated using PGD-10 with an attack strength of 1/255. Additionally, we also report $A_{last}$ on adversarial examples generated by Auto. with the same attack strength.

**Experiments on Short Tasks.** Table 1 shows that directly applying adversarial training in class-incremental learning leads to severe forgetting, with $BWT$ values dropping below -90%, and also fails to provide satisfactory adversarial robustness. Among the compared methods, R-LwF-MC and FLAIR rely on a binary cross-entropy loss that treats each class independently, which results in second-best performance. Our method further improves adversarial robustness on top of these approaches, achieving significant gains. For example, compared to FLAIR on S-CIFAR10, our method improves PGD $A_{last}$ by 15.54%. However, it is worth noting that the clean accuracy of R-LwF-MC, FLAIR, and our method is lower than that of FARE. This is because FARE focuses on protecting clean accuracy first and only then improving adversarial robustness.

**Experiments on Long Tasks.** To further compare the performance of different methods, we conduct experiments on S-CIFAR100 and S-TinyImageNet under the 10-task setting. As shown in Table 2, SAGE achieves the second-best performance on clean examples across both datasets and is comparable to FARE. On adversarial examples, however, SAGE demonstrates clear advantages. On S-CIFAR100, it surpasses the second-best method R-SG by 16.39% in PGD $A_{last}$, while on S-TinyImageNet it exceeds FARE by 11.10%. It is also noteworthy that methods such as FLAIR and R-LwF-MC, which perform well in short-task settings, show poor performance in this long-task setting, likely due to their shared reliance on binary cross-entropy (BCE) loss. Prior work shows that, unlike cross-entropy, BCE lacks normalization and inter-class competition, leading to weaker separability on large-scale datasets (Li et al., 2025). Consistent with this, our experiments reveal that BCE can cause output collapse, where the model predicts a single dominant class.

## 5.3 ABLATION STUDIES

**Impact of Adversarial Attack Strength.** The model is adversarially trained with a perturbation strength of 1/255. To further assess its robustness, we evaluate it under stronger perturbations of 2/255 and 4/255, and report the averaged results across all settings. As shown in Table 3, although the performance of all methods decreases as the attack strength increases, SAGE consistently outperforms all baselines across different datasets. Notably, on the smaller S-STL10 dataset, SAGE shows a modest 2.65% improvement over the second-best method in PGD $A_{last}$. In contrast, on the larger S-TinyImageNet dataset, which has more classes and a longer task sequence, making it more

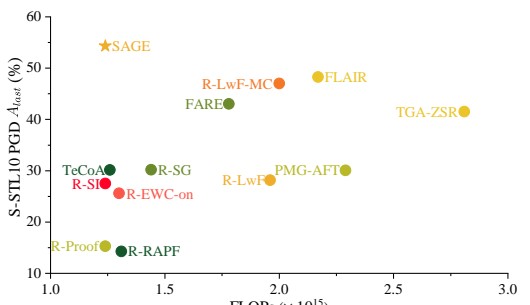 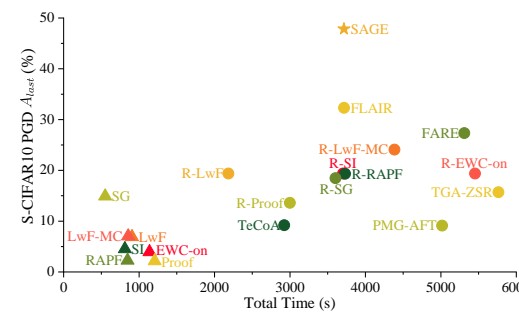

(a) Comparison of Robustness and Training FLOPs across different methods on S-STL10.

(b) Comparison of Robustness and Total Time across different methods on S-CIFAR10.

Figure 2: Comparison of computational cost (FLOPs) and total training time across different methods. ▲ represents standard CIL approaches, ● denotes AT-based methods, and our method is highlighted with ★ for clear visual distinction in the comparison.

difficult to retain prior knowledge and resist adversarial perturbations, SAGE achieves a substantial 9.77% improvement. These gains highlight that SAGE not only maintains robustness under mild perturbations but also demonstrates superior resilience as attack strength increases.

**Module Ablation.** To comprehensively evaluate the effectiveness of SAGE, we conduct ablation studies on both the contrastive loss formulations and the parameter importance strategies. As shown in Table 4a, using $L_{con}(\boldsymbol{a}, \boldsymbol{c})^1$ and $L_{con}(\boldsymbol{b}, \boldsymbol{c})$ individually causes the model to focus on only one aspect, leading to suboptimal performance. However, when combined, these losses enable the model to improve both adversarial robustness and resistance to catastrophic forgetting, leading to gains across multiple dimensions. Achieving the right balance between these two remains challenging. Our proposed $L_{con}(\boldsymbol{a}, \boldsymbol{b})$ resolves this issue by eliminating the need to explicitly manage this trade-off, and it outperforms the combined use of $L_{con}(\boldsymbol{a}, \boldsymbol{c})$ and $L_{con}(\boldsymbol{b}, \boldsymbol{c})$, further demonstrating its superiority. Additionally, as shown in Table 4b, the selective parameter optimization strategy can improve the model's clean accuracy to some extent. However, simpler parameter importance evaluation methods often lead to degraded performance, as demonstrated by a 9.72% drop in PGD $A_{last}$ performance when using $|\theta^l|$ on S-CIFAR100. This underscores the need for a more robust evaluation approach, and the method we propose delivers improvements across multiple datasets.

For completeness, we present supplementary ablation experiments in the Appendix E, specifically analyzing the effect of top-$k\%$ parameter selection and the sensitivity of the loss weight $\mu$.

## 5.4 EFFICIENCY OF COMPUTATION AND TRAINING TIME

Adversarial Training introduces a significant training-time overhead relative to standard CIL baselines that operate only on clean examples, calling its scalability into question for large incremental tasks. To address this concern, we carried out experiments specifically designed to assess computational efficiency and training time.

**Comparison of Computational Cost (FLOPs).** We compare the robustness and training FLOPs of our method with representative benchmark methods on S-STL10, as shown in Figure 2a. The results demonstrate that our method achieves consistently higher robustness under adversarial attacks while requiring substantially fewer training FLOPs than other baselines. In particular, SAGE achieves a 42.90% reduction in computational overhead compared to FLAIR, a strong baseline in the RCIL, while further enhancing robustness. These results highlight the effectiveness of SAGE, which not only strengthens robustness but also improves training efficiency, thus providing a favorable robustness-efficiency trade-off for class-incremental learning. Such a balance is especially important in practical class-incremental learning settings, where limited computational resources and robustness to adversarial attacks are critical requirements.

---

[1]To simplify the notation, we use the definition from Eq. 14.



Figure 3: Confusion matrices for clean and adversarial examples on S-CIFAR10. The horizontal axis shows predicted classes, and the vertical axis shows ground-truth classes. Brighter diagonal values indicate higher classification accuracy, whereas off-diagonal values correspond to misclassifications.

**Comparison of Total Time.** We further compare the robustness and total time of our method with three categories of approaches: standard CIL baselines trained only on clean data, adversarially trained baselines (AT, R-CIL, and R-CIL-CLIP), and a robust class-incremental learning method. For standard CIL baselines, we adopt partial fine-tuning, where all methods update only the last block, except RAPF and Proof, which also update the added adapter module. To ensure adversarial robustness, all AT-based methods and RCIL use full fine-tuning. The total time (in seconds) for the entire training and validation process on S-CIFAR10 is reported in Figure 2b. The experiments show that standard CIL baselines achieve the fastest training, but at the cost of limited adversarial robustness. In comparison, SAGE operates within a similar computational budget while consistently providing notably stronger robustness.

## 5.5 VISUALIZATION OF CLEAN AND ADVERSARIAL CONFUSION MATRICES

Figure 3 provides further insights into the superior performance of our method SAGE. The baseline method exhibits a noticeable drop in performance, as evidenced by the reduced brightness along the diagonal, and tends to suffer from catastrophic forgetting, where predictions are biased toward the newly learned classes. This leads to outputs concentrated on the right side of the confusion matrix. In contrast, SAGE not only preserves strong classification accuracy on clean examples and achieves a balanced plasticity-stability trade-off, but also sustains robustness against adversarial perturbations. The confusion matrices clearly show that our approach alleviates class bias, maintains brighter diagonal patterns, and distributes predictions more uniformly across both old and new tasks, highlighting its effectiveness in both clean and adversarial settings.

In addition, we also provide the experimental results with replay in Appendix F, the performance on S-ImageNet-A in Appendix G, a comparison with free adversarial training in Appendix H, and an evaluation using a robust backbone initialization in Appendix I.

## 6 CONCLUSIONS

Although CIL mitigates catastrophic forgetting, its susceptibility to adversarial perturbations limits practical applicability. Existing RCIL methods attempt to address this challenge but remain inadequate due to simplistic designs. To overcome these limitations, we introduce SAGE, which advances RCIL by integrating a selective parameter optimization scheme for adversarial training with a geometry-constrained contrastive loss, thereby improving adversarial robustness while mitigating forgetting. Extensive experiments show that SAGE not only surpasses a naive combination of CIL and AT but also consistently outperforms prior RCIL methods, yielding significant improvements across multiple datasets.

**Limitations and Future Work.** While SAGE achieves notable gains, it still depends on computationally intensive adversarial training, which may hinder scalability to larger models or datasets. In addition, the joint challenge of mitigating forgetting while enhancing adversarial robustness remains significant, leaving substantial room for further improvement. Future work will explore more efficient robustness techniques to reduce computational overhead and close the performance gap between RCIL and existing CIL methods.

**Ethics Statement.** This work does not involve human subjects, sensitive personal data, or experiments that could raise ethical concerns. All datasets used are publicly available and widely adopted in prior research. Our methodology focuses on improving robustness and knowledge retention in class-incremental learning, without introducing potential risks of harmful applications. In particular, by enhancing adversarial robustness, our approach contributes to improving the security and reliability of models in real-world applications. Finally, we have adhered to the Code of Ethics throughout the research and submission process.

**Reproducibility Statement.** We have made significant efforts to ensure the reproducibility of our work. In Section 5.1, we describe the datasets, experimental settings, training configurations, and hyperparameters. We further provide detailed training procedures for all baseline methods in Appendix J, and we also include our code in the supplementary material.

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

## A    LLM USAGE STATEMENT

We used a large language model, ChatGPT, as a general-purpose assistant to help with language polishing and stylistic improvements of the manuscript. The LLM was not involved in research ideation, experimental design, data analysis, or interpretation of results, and all scientific contributions and conclusions presented in this work are solely those of the authors.

## B    EXPERIMENTAL VALIDATION OF THE TRADE-OFF IN RCIL

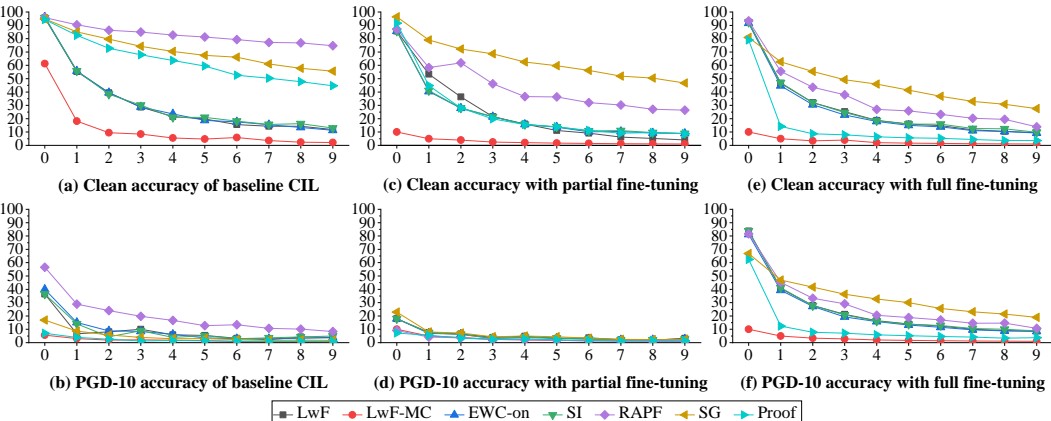

Figure 4: Incremental performance (clean and PGD-10 accuracy) of different methods in the CIL setting on S-CIFAR100. **(a)** and **(b)** show the clean and PGD-10 accuracy of baseline CIL. **(c)** and **(d)** show the clean and PGD-10 accuracy of CIL with adversarial training (AT) using partial fine-tuning (all methods update only the last block, except RAPF and Proof that additionally update the added adapter module). **(e)** and **(f)** show the clean and PGD-10 accuracy of CIL with AT using full fine-tuning.

Most existing CIL methods focus on alleviating catastrophic forgetting, but they largely overlook the vulnerability of CIL models to adversarial perturbations. Such vulnerability poses a critical threat to the reliability and safety of CIL in real-world applications. This trade-off is formally captured in Eq. **??**, where prior work tends to emphasize the first term, while enforcing stability for past tasks, which ignores the second term, which promotes robustness against adversarial perturbations. As a result, models that achieve high accuracy on clean examples often collapse under adversarial ones, producing incorrect or misleading predictions and leading to nearly zero accuracy (see Figure 4 (a, b)). Incorporating adversarial training into CIL provides a natural way to address this issue by encouraging models to balance robustness and adaptive learning across tasks. However, striking this balance remains highly challenging, as robustness and stability impose conflicting requirements on parameter updates. For example, Figure 4 (c, d) shows that updating only a limited set of parameters yields minimal robustness gains with less forgetting, whereas Figure 4 (e, f) shows that updating all parameters improves robustness at the cost of severe forgetting.

These observations highlight the fundamental difficulty of designing RCIL algorithms that can simultaneously retain past knowledge and defend against adversarial perturbations, underscoring the necessity of new approaches that explicitly address this robustness-stability trade-off.

## C    DETAILED PROOF OF THE GEOMETRIC INEQUALITY

In particular, for any three unit vectors $a$, $b$, and $c$, the vector $a$ can be orthogonally decomposed into two components: one lying in the direction of $c$, and the other residing in the subspace orthogonal to $c$. Formally,

$$a = (a^\top c)\,c + a_\perp, \quad \text{where } a_\perp \perp c \text{ and thus } a_\perp^\top c = 0 \tag{17}$$

Similarly, the vector $b$ can be decomposed as:

$$b = (b^\top c)\,c + b_\perp, \quad \text{where } b_\perp \perp c \text{ and thus } b_\perp^\top c = 0 \tag{18}$$

Thus, the inner product $\boldsymbol{a}^\top \boldsymbol{b}$ becomes:

$$
\begin{aligned}
\boldsymbol{a}^\top \boldsymbol{b} &= ((\boldsymbol{a}^\top \boldsymbol{c})\,\boldsymbol{c} + \boldsymbol{a}_\perp)^\top ((\boldsymbol{b}^\top \boldsymbol{c})\,\boldsymbol{c} + \boldsymbol{b}_\perp) \\
&= (\boldsymbol{a}^\top \boldsymbol{c})(\boldsymbol{b}^\top \boldsymbol{c})\boldsymbol{c}^\top \boldsymbol{c} + (\boldsymbol{a}^\top \boldsymbol{c})\boldsymbol{c}^\top \boldsymbol{b}_\perp + (\boldsymbol{a}^\top \boldsymbol{c})\boldsymbol{c}_\perp^\top \boldsymbol{c} + \boldsymbol{a}_\perp^\top \boldsymbol{b}_\perp \\
&= (\boldsymbol{a}^\top \boldsymbol{c})(\boldsymbol{b}^\top \boldsymbol{c}) + \boldsymbol{a}_\perp^\top \boldsymbol{b}_\perp
\end{aligned}
\tag{19}
$$

Therefore, the deviation from the product of cosine similarities is:

$$
\begin{aligned}
\left|\boldsymbol{a}^\top \boldsymbol{b} - (\boldsymbol{a}^\top \boldsymbol{c})(\boldsymbol{b}^\top \boldsymbol{c})\right| &= |\boldsymbol{a}_\perp^\top \boldsymbol{b}_\perp| \\
&\le \|\boldsymbol{a}_\perp\| \cdot \|\boldsymbol{b}_\perp\| \\
&= \left\|\boldsymbol{a} - (\boldsymbol{a}^\top \boldsymbol{c})\,\boldsymbol{c}\right\| \cdot \left\|\boldsymbol{b} - (\boldsymbol{b}^\top \boldsymbol{c})\,\boldsymbol{c}\right\|
\end{aligned}
\tag{20}
$$

Because they are unit vectors, we have:

$$
\left\|\boldsymbol{a} - (\boldsymbol{a}^\top \boldsymbol{c})\,\boldsymbol{c}\right\|^2 = \boldsymbol{a}^\top \boldsymbol{a} - 2(\boldsymbol{a}^\top \boldsymbol{c})^2 + (\boldsymbol{a}^\top \boldsymbol{c})^2\, \boldsymbol{c}^\top \boldsymbol{c} = 1 - (\boldsymbol{a}^\top \boldsymbol{c})^2
\tag{21}
$$

Combining these results yields:

$$
\left|\boldsymbol{a}^\top \boldsymbol{b} - (\boldsymbol{a}^\top \boldsymbol{c})(\boldsymbol{b}^\top \boldsymbol{c})\right| \le \left\|\boldsymbol{a} - (\boldsymbol{a}^\top \boldsymbol{c})\,\boldsymbol{c}\right\| \cdot \left\|\boldsymbol{b} - (\boldsymbol{b}^\top \boldsymbol{c})\,\boldsymbol{c}\right\| = \sqrt{1 - (\boldsymbol{a}^\top \boldsymbol{c})^2} \cdot \sqrt{1 - (\boldsymbol{b}^\top \boldsymbol{c})^2}
\tag{22}
$$

Because $\gamma_{\boldsymbol{ab}} = \cos(\boldsymbol{a}, \boldsymbol{b}) = \boldsymbol{a}^\top \boldsymbol{b}$, $\gamma_{\boldsymbol{ac}} = \cos(\boldsymbol{a}, \boldsymbol{c}) = \boldsymbol{a}^\top \boldsymbol{c}$, and $\gamma_{\boldsymbol{bc}} = \cos(\boldsymbol{b}, \boldsymbol{c}) = \boldsymbol{b}^\top \boldsymbol{c}$, we obtain:

$$
|\gamma_{\boldsymbol{ab}} - \gamma_{\boldsymbol{ac}}\gamma_{\boldsymbol{bc}}| \le \sqrt{1 - \gamma_{\boldsymbol{ac}}^2} \cdot \sqrt{1 - \gamma_{\boldsymbol{bc}}^2}
\tag{23}
$$

This implies the final bound:

$$
\gamma_{\boldsymbol{ac}}\gamma_{\boldsymbol{bc}} - \sqrt{(1 - \gamma_{\boldsymbol{ac}}^2)(1 - \gamma_{\boldsymbol{bc}}^2)} \le \gamma_{\boldsymbol{ab}} \le \gamma_{\boldsymbol{ac}}\gamma_{\boldsymbol{bc}} + \sqrt{(1 - \gamma_{\boldsymbol{ac}}^2)(1 - \gamma_{\boldsymbol{bc}}^2)}
\tag{24}
$$

# D  PSEUDO CODE FOR SAGE

To provide a clearer exposition of our method, we present the corresponding pseudo code in Algorithm 1.

---

**Algorithm 1** PSEUDO CODE FOR SAGE.

---

**Input:** Incremental Datasets: $\{\mathcal{D}_1, \mathcal{D}_2, \cdots, \mathcal{D}_T\}$, Pre-trained CLIP image encoder and text encoder: $f(\cdot), g(\cdot)$
**Output:** Robust incrementally trained CLIP model

1: **for** $t = 1$ **to** $T$ **do**
2:     Extract input samples $\mathcal{X}_t$, corresponding labels $\mathcal{Y}_t$, and text prompts $\mathcal{P}_t$ from $\mathcal{D}_t$
3:     **for** epoch = 1 **to** max_epochs **do**
4:         Sample a mini-batch: input samples $\boldsymbol{x}_t$, corresponding labels $\boldsymbol{y}_t$, and text prompts $\boldsymbol{p}_t$
5:         **for** iter = 1 **to** max_iterations **do**
6:             Generate adversarial examples $\boldsymbol{x}_t^{adv}$ using PGD via Eq. 1
7:         **end for**
8:         **if** $t > 1$ **then**
9:             Compute the total loss $\mathcal{L}_{\text{total}}$ via Eq. 16
10:        **else**
11:            Compute the cross-entropy loss $\mathcal{L}_{\text{CE}}$
12:        **end if**
13:        Perform backpropagation to compute gradients $\frac{\partial \mathcal{L}}{\partial \theta}$
14:        Compute parameter importance scores $I$
15:        Determine the binary mask $m$ via Eq. 10
16:        Update the parameters of the image encoder via $\theta \leftarrow \theta + m \odot \Delta\theta$
17:     **end for**
18: **end for**

---

## E  SUPPLEMENTARY ABLATION EXAMPLES

**Effect of the Top-$k\%$ Parameter Selection.** We update the top-$k\%$ of parameters in each layer based on their importance score $I$. To assess the impact of different $k$ values on the final performance, we conduct experiments with $k = 1e-3, 1e-2, 1e-1$. The results in Table 5 show that when $k$ is small (e.g., $1e-3$), the model tends to exhibit weaker robustness.

Table 5: Effect of the Top-$k\%$ parameter selection. **Bold** for the best result.

| Top-$k\%$ | S-CIFAR10 | | | S-CIFAR100 | | |
|---|---|---|---|---|---|---|
| | Clean $\overline{A}$ | PGD $\overline{A}$ | Auto. $A_{last}$ | Clean $\overline{A}$ | PGD $\overline{A}$ | Auto. $A_{last}$ |
| $1e-3$ | **72.93** | 60.57 | 41.32 | 59.83 | 44.70 | 25.70 |
| $1e-2$ | 72.36 | **61.75** | **41.60** | **63.20** | **48.49** | 28.98 |
| $1e-1$ | 70.77 | 61.11 | 40.07 | 61.68 | 47.99 | **29.09** |

This effect is not obvious on datasets with fewer classes, but on S-CIFAR100, the PGD $\overline{A}$ score decreases by 3.79% compared to $k = 1e-2$. On the other hand, when $k$ is large (e.g., $1e-1$), the generalization performance is worse than that of $k = 1e-2$. This observation is consistent with our discussion in Section 4.1, where class-incremental learning tends to update fewer parameters to mitigate forgetting, while adversarial robustness benefits from updating a larger portion of parameters to enforce robustness. Considering the trade-off, we set $k = 1e-2$.

**Sensitivity of the Loss Weight $\mu$.** Figure 5 illustrates the sensitivity of the weight $\mu$ on S-CIFAR10 and S-CIFAR100. The results indicate that the model exhibits sharp performance degradation when $\mu < 0.75$, indicating a high sensitivity to under-weighting the loss term. For instance, on S-CIFAR10, the PGD $A_{last}$ drops sharply from 47.85% at $\mu = 1$ to 19.31% at $\mu = 0$. Once $\mu > 0.75$, both clean and robust accuracies become relatively stable, with values around $\mu = 1$ achieving the best trade-off. Overall, extreme values of $\mu$ substantially impair performance, whereas settings close to $\mu = 1$ yield stable and optimal results. Hence, we choose $\mu = 1$ as the default setting.

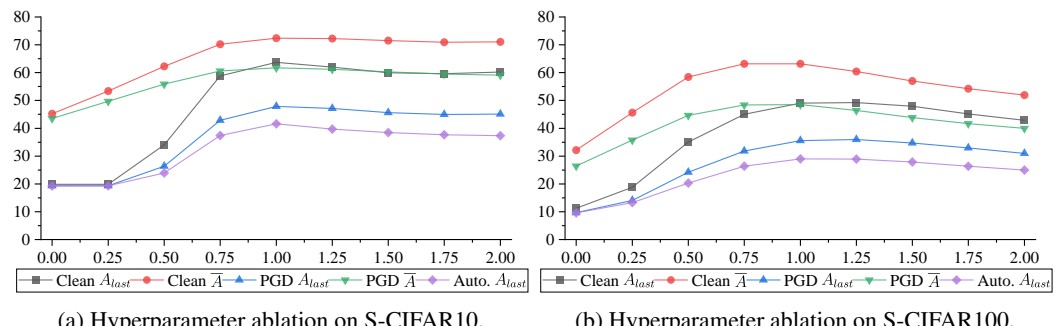

(a) Hyperparameter ablation on S-CIFAR10.          (b) Hyperparameter ablation on S-CIFAR100.

Figure 5: Effect of varying the loss weight $\mu$ on performance for S-CIFAR10 and S-CIFAR100 after fine-tuning with PGD-2: (a) S-CIFAR10; (b) S-CIFAR100.

## F  EXPERIMENTAL RESULTS WITH REPLAY

In addition to the above methods, we also compare with TABA (Bai et al., 2023), a replay-based RCIL approach. To enable replay, we adopt iCaRL (Rebuffi et al., 2017) to store 500 exemplars from previous tasks, which improves the model's performance and allows evaluation under the exemplar-replay setting. As shown in Table 6, all methods achieve performance gains over the no-replay setting. Across all four datasets, our method SAGE consistently outperforms the others in terms of PGD $A_{last}$, yielding the smallest improvement of 3.26% on S-STL10 and the largest improvement of 18.40% on S-CIFAR100 compared to the second-best approach. Meanwhile, our method also achieves the best results on Auto. $A_{last}$. This further demonstrates the effectiveness of our approach, showing that it maintains strong performance in both replay and no-replay settings.

## G  EXPERIMENTAL RESULTS ON IMAGENET-A

To evaluate the performance of our method SAGE in realistic continual learning scenarios, we extend our experiments to ImageNet-A Hendrycks et al. (2021). This dataset consists of natural images that are particularly challenging for standard ImageNet-trained models, selected to highlight common

Table 6: Evaluation of several methods on ViT-B/32 with 500 size of memory buffer. We report average PGD-10, Auto. accuracy (%) and $BWT$ on S-CIFAR10, S-STL10, S-CIFAR100 and S-TinyImageNet under attack strength of 1/255. **Bold** for the best result, underline for secondary.

| Type | Method | S-CIFAR10 | | | | S-STL10 | | | | S-CIFAR100 | | | | S-TinyImageNet | | | |
|---|---|---|---|---|---|---|---|---|---|---|---|---|---|---|---|---|---|
| | | PGD | | | Auto. | PGD | | | Auto. | PGD | | | Auto. | PGD | | | Auto. |
| | | $\overline{A}\uparrow$ | $A_{last}\uparrow$ | $BWT\uparrow$ | $A_{last}\uparrow$ | $\overline{A}\uparrow$ | $A_{last}\uparrow$ | $BWT\uparrow$ | $A_{last}\uparrow$ | $\overline{A}\uparrow$ | $A_{last}\uparrow$ | $BWT\uparrow$ | $A_{last}\uparrow$ | $\overline{A}\uparrow$ | $A_{last}\uparrow$ | $BWT\uparrow$ | $A_{last}\uparrow$ |
| AT | TeCoA | 18.05 | 9.42 | -41.56 | 0.13 | 44.65 | 35.72 | -24.34 | 0.36 | 13.71 | 7.99 | -47.90 | 6.83 | 12.75 | 5.66 | -40.23 | 4.45 |
| | FARE | 38.64 | 30.88 | -10.76 | 3.69 | 57.91 | 49.85 | -8.70 | 9.78 | 25.66 | 18.97 | -6.10 | 5.72 | 24.80 | 22.63 | -5.70 | 11.36 |
| | PMG-AFT | 18.16 | 9.07 | -42.13 | 0.05 | 44.07 | 35.45 | -24.09 | 0.31 | 13.89 | 8.31 | -47.94 | 7.18 | 13.27 | 5.88 | -40.88 | 4.65 |
| | TGA-ZSR | 38.67 | 19.28 | -63.45 | 13.55 | 62.55 | 54.30 | -17.44 | 26.10 | 25.72 | 13.71 | -48.67 | 8.86 | 17.22 | 8.53 | -36.56 | 5.34 |
| R-CIL | R-LwF | 72.98 | 58.04 | -46.46 | 57.51 | 81.08 | 71.34 | -23.11 | 69.55 | 41.27 | 21.02 | -67.76 | 20.79 | 23.85 | 8.76 | -56.19 | 8.62 |
| | R-LwF-MC | 56.41 | 34.42 | -73.58 | 31.50 | 55.03 | 29.50 | -68.11 | 16.89 | 2.93 | 1.00 | -1.11 | 1.00 | 1.46 | 0.50 | -0.00 | 0.50 |
| | R-EWC-on | 72.38 | 57.82 | -46.61 | 57.55 | 79.01 | 67.14 | -28.59 | 66.67 | 37.28 | 17.00 | -68.97 | 16.88 | 21.07 | 8.23 | -52.46 | 8.13 |
| | R-SI | 74.11 | 59.05 | -45.25 | 58.53 | 80.33 | 70.58 | -25.55 | 68.61 | 42.45 | 20.63 | -68.42 | 20.46 | 24.99 | 10.56 | -55.61 | 10.38 |
| R-CIL-CLIP | R-RAPF | 72.20 | 56.49 | -49.13 | 0.00 | 59.52 | 62.92 | -25.77 | 61.94 | 46.34 | 26.17 | -62.80 | 25.69 | 34.28 | 21.85 | -51.34 | 21.34 |
| | R-SG | 55.64 | 50.57 | 2.40 | 0.00 | 76.32 | 70.75 | -16.23 | 62.84 | 39.24 | 24.56 | -32.93 | 20.04 | 25.38 | 14.09 | -20.74 | 11.11 |
| | R-Proof | 33.79 | 13.87 | -69.00 | 13.14 | 64.70 | 33.98 | -61.52 | 32.10 | 13.31 | 4.83 | -41.10 | 4.42 | 5.02 | 1.97 | -12.82 | 1.49 |
| RCIL | TABA | 62.39 | 40.34 | -64.36 | 36.52 | 62.73 | 37.91 | -34.42 | 22.12 | 2.94 | 1.00 | -1.11 | 1.00 | 1.46 | 0.50 | -0.00 | 0.50 |
| | FLAIR | 58.10 | 40.52 | -67.11 | 35.59 | 62.95 | 47.78 | -49.89 | 26.41 | 2.93 | 1.00 | -1.11 | 1.00 | 1.46 | 0.50 | -0.00 | 0.50 |
| RCIL4CLIP | SAGE (ours) | **75.09** | **66.26** | -13.15 | **58.82** | **81.30** | **74.60** | **-1.33** | **72.19** | **55.36** | **44.57** | -1.67 | **38.42** | **43.57** | **37.14** | -1.19 | **31.30** |

failure cases. The images exhibit diverse and complex visual conditions such as unusual viewpoints, occlusions, lighting variations, blur, and background clutter. Using this dataset allows us to assess how well our method handles the challenges posed by real-world, diverse, and hard-to-classify images. ImageNet-A is divided into 10 tasks with 20 classes per task, denoted as S-ImageNet-A. The relatively limited sample size further increases the difficulty, since each class contains far fewer examples than conventional large-scale benchmarks, making continual learning more sensitive to overfitting and feature drift. The results in Table 7 show that SAGE consistently delivers higher robustness and lower forgetting on S-ImageNet-A, remaining stable even under severe distribution shifts and limited samples, demonstrating its effectiveness in realistic continual learning settings.

Table 7: Evaluation of several methods on ViT-B/32 without memory. We report Clean, PGD-10, Auto. accuracy (%), and $BWT$ on S-ImageNet-A under attack strength of 1/255. **Bold** for the best result, underline for secondary.

| Type | Method | S-ImageNet-A | | | | | | |
|---|---|---|---|---|---|---|---|---|
| | | Clean | | | PGD | | | Auto. |
| | | $\overline{A}\uparrow$ | $A_{last}\uparrow$ | $BWT\uparrow$ | $\overline{A}\uparrow$ | $A_{last}\uparrow$ | $BWT\uparrow$ | $A_{last}\uparrow$ |
| AT | TeCoA | 12.16 | 2.40 | -24.68 | 0.81 | 1.63 | -3.72 | 0.07 |
| | FARE | 24.53 | 12.82 | -15.50 | 2.13 | 1.53 | -0.96 | 0.26 |
| | PMG-AFT | 12.39 | 2.40 | -25.64 | 0.89 | 1.63 | -3.98 | 0.07 |
| | TGA-ZSR | 13.69 | 3.50 | -29.26 | 2.54 | 2.42 | -8.18 | 0.86 |
| R-CIL | R-LwF | 7.48 | 3.75 | -22.70 | 3.53 | 2.21 | -11.60 | 1.51 |
| | R-LwF-MC | 1.14 | 0.39 | -0.00 | 1.14 | 0.39 | -0.00 | 0.79 |
| | R-EWC-on | 7.72 | 3.27 | -23.06 | 3.71 | 2.60 | -11.51 | 1.71 |
| | R-SI | 7.54 | 2.98 | -22.11 | 3.46 | 2.21 | -10.52 | 1.51 |
| R-CIL-CLIP | R-RAPF | 2.22 | 1.92 | -8.38 | 1.89 | 1.63 | -6.53 | 1.05 |
| | R-SG | 5.36 | 3.01 | -5.60 | 2.45 | 1.51 | -1.89 | 0.92 |
| | R-Proof | 5.13 | 1.63 | -12.78 | 2.82 | 1.54 | -8.12 | 0.59 |
| RCIL | FLAIR | 1.14 | 0.39 | -0.00 | 1.14 | 0.39 | -0.00 | 0.79 |
| RCIL4CLIP | SAGE (ours) | 20.45 | **13.81** | -8.61 | **8.91** | **6.69** | -3.38 | **4.94** |

## H COMPARISON WITH FREE ADVERSARIAL TRAINING

To strengthen the assessment of practicality, we additionally include Free Adversarial Training (FreeAT) Shafahi et al. (2019a) as a lightweight robustness baseline. FreeAT avoids generating adversarial examples in separate steps and instead updates the model and the adversarial perturbation within the same backward pass. This design greatly reduces the computational overhead during training. For FreeAT, we set the nominal number of epochs to be consistent with those used for the other methods in order to ensure a fair comparison. Because FreeAT performs $m$ gradient update hops within each epoch, its effective number of passes over the data becomes epochs divided by $m$. In our experiments, we use $m = 2$, which keeps the computational cost comparable across methods while still following the update mechanism required by FreeAT. The results in Table 8 show that across all sequential tasks and both attacks, SAGE consistently and significantly outperforms FreeAT, delivering stronger robustness and much less forgetting, making it a more suitable choice for robust class-incremental learning.

Table 8: Evaluation of several methods on ViT-B/32 without memory. We report Clean, PGD-10, Auto. accuracy (%), and $BWT$ on S-CIFAR10, S-STL10, S-CIFAR100, and S-TinyImageNet under attack strength of 1/255. **Bold** for the best result.

| Dataset | Method | Clean | | | PGD | | | Auto. |
|---|---|---|---|---|---|---|---|---|
| | | $\overline{A}\uparrow$ | $A_{last}\uparrow$ | $BWT\uparrow$ | $\overline{A}\uparrow$ | $A_{last}\uparrow$ | $BWT\uparrow$ | $A_{last}\uparrow$ |
| S-CIFAR10 | FreeAT | 44.85 | 19.78 | -98.23 | 41.90 | 19.15 | -90.81 | 18.03 |
| | SAGE (ours) | **72.36** | **63.67** | **-34.12** | **61.75** | **47.85** | **-42.27** | **41.60** |
| S-STL10 | FreeAT | 54.66 | 39.81 | -73.77 | 44.63 | 24.49 | -80.11 | 17.74 |
| | SAGE (ours) | **73.52** | **69.56** | **-33.16** | **63.96** | **54.31** | **-40.20** | **46.32** |
| S-CIFAR100 | FreeAT | 26.68 | 9.38 | -89.61 | 20.85 | 7.78 | -69.96 | 7.36 |
| | SAGE (ours) | **63.20** | **49.02** | **-22.62** | **48.49** | **35.59** | **-19.89** | **28.98** |
| S-TinyImageNet | FreeAT | 21.36 | 7.65 | -70.81 | 12.71 | 4.37 | -43.34 | 4.23 |
| | SAGE (ours) | **56.14** | **44.72** | **-13.54** | **40.21** | **31.95** | **-9.49** | **26.18** |

## I EVALUATING PERFORMANCE WITH ROBUST BACKBONE INITIALIZATION

To assess whether a robustly fine-tuned backbone can further enhance our approach, we initialized the model using a FARE Schlarmann et al. (2024) pretrained Tiny-ImageNet checkpoint. As shown in Table 9, this initialization leads to consistent improvements on S-CIFAR10, S-STL10, and S-TinyImageNet, while yielding a small drop on S-CIFAR100. The overall trend shows that robust initialization helps SAGE adapt more confidently to most continual learning scenarios and encourages a more stable progression of robustness.

Table 9: Evaluation of several methods on ViT-B/32 without memory. We report Clean, PGD-10, Auto. accuracy (%), and $BWT$ on S-CIFAR10, S-STL10, S-CIFAR100, and S-TinyImageNet under attack strength of 1/255. **Bold** for the best result.

| Dataset | Method | Clean | | | PGD | | | Auto. |
|---|---|---|---|---|---|---|---|---|
| | | $\overline{A}$↑ | $A_{last}$↑ | $BWT$↑ | $\overline{A}$↑ | $A_{last}$↑ | $BWT$↑ | $A_{last}$↑ |
| S-CIFAR10 | FLAIR | 61.27 | 45.83 | -66.05 | 51.73 | 32.31 | -77.91 | 30.90 |
| | FLAIR + FARE | 48.15 | 25.66 | -91.76 | 44.34 | 20.26 | -94.80 | 19.51 |
| | SAGE | 72.36 | 63.67 | **-34.12** | 61.75 | 47.85 | **-42.27** | 41.60 |
| | SAGE + FARE | **73.52** | **66.86** | -34.24 | **62.22** | **50.33** | -45.00 | **45.52** |
| S-STL10 | FLAIR | 71.32 | 64.92 | -42.17 | 59.69 | 48.26 | -51.59 | 41.65 |
| | FLAIR + FARE | 79.21 | 81.11 | -22.44 | 62.81 | 51.39 | -54.02 | 42.76 |
| | SAGE | 73.52 | 69.56 | -33.16 | 63.96 | 54.31 | -40.20 | 46.32 |
| | SAGE + FARE | **80.07** | **82.42** | **-17.97** | **70.91** | **66.75** | **-30.97** | **60.59** |
| S-CIFAR100 | FLAIR | 3.05 | 1.00 | **-0.00** | 2.94 | 1.00 | **-0.00** | 1.00 |
| | FLAIR + FARE | 3.15 | 1.00 | -1.18 | 3.03 | 1.00 | -1.12 | 1.00 |
| | SAGE | **63.20** | **49.02** | -22.62 | **48.49** | **35.59** | -19.89 | **28.98** |
| | SAGE + FARE | 60.07 | 44.65 | -35.89 | 46.00 | 30.75 | -34.48 | 24.76 |
| S-TinyImageNet | FLAIR | 1.46 | 0.50 | **-0.00** | 1.46 | 0.50 | **-0.00** | 0.50 |
| | FLAIR + FARE | 1.46 | 0.50 | **-0.00** | 1.46 | 0.50 | **-0.00** | 0.50 |
| | SAGE | 56.14 | 44.72 | -13.54 | 40.21 | 31.95 | -9.49 | 26.18 |
| | SAGE + FARE | **61.03** | **52.74** | -23.52 | **43.59** | **36.31** | -21.98 | **30.51** |

## J TRAINING DETAILS OF BASELINES

CLIP (Radford et al., 2021a) consists of an image encoder $f(\cdot)$ and a text encoder $g(\cdot)$, which maps a given image-text pair $(\boldsymbol{x}, \boldsymbol{p})$ into corresponding image embedding $f(\boldsymbol{x})$ and text embedding $g(\boldsymbol{p})$. The prediction for the correspondence between image $\boldsymbol{x}$ and $\boldsymbol{p}$ is computed as:

$$\mathcal{S}_{i,j} = \tau \cdot \frac{f(\boldsymbol{x}_i)}{\|f(\boldsymbol{x}_i)\|} \cdot \left(\frac{g(\boldsymbol{p}_j)}{\|g(\boldsymbol{p}_j)\|}\right)^{\top}$$
$$\boldsymbol{q}_{i,j} = \frac{\mathcal{S}_{i,j}}{\sum_k \mathcal{S}_{i,k}} \tag{25}$$

where $i$ and $j$ index the image and text samples respectively, and $\tau$ is a temperature parameter that scales the similarity scores. Here, we denote by $\mathcal{S}^{adv}$ the output generated from adversarial examples $\boldsymbol{x}^{adv}$ and by $\boldsymbol{q}^{adv}$ the corresponding predictions, while $\mathcal{S}_t$ represents the output after learning the $t$-th task. Furthermore, $\mathcal{S}_t|_i^j$ denotes the output restricted to the classes introduced from task $i$ through task $j$, after completing the training of the current task $t$. In this work, we update all model parameters during training, adopt the SGD optimizer to minimize the objective function, and employ the text prompt template "This is a photo of {}".

### J.1 ZERO-SHOT ADVERSARIAL ROBUSTNESS

#### J.1.1 THE TRAINING DETAILS OF TeCoA

TeCoA (Mao et al., 2023) is a simple yet effective approach to enhance adversarial robustness. It introduces a text-guided contrastive adversarial training loss, which enforces alignment between the adversarial visual features and their corresponding text embeddings. Thus, the loss is defined as:

$$\mathcal{L}_{\text{total}} = \mathcal{L}_{\text{CE}}(\boldsymbol{q}_t^{adv}, \boldsymbol{y}) \tag{26}$$

where $\boldsymbol{q}_t^{adv}$ denotes the prediction vector of adversarial examples from task $t$, and $\boldsymbol{y}$ represents the one-hot vector label.

For S-CIFAR10 and S-STL10, the model is trained for 20 epochs with a learning rate of 0.001, a weight decay of 1e-5, and a batch size of 64. For S-CIFAR100 and S-TinyImageNet, we use the same hyperparameters but extend the training to 50 epochs.

#### J.1.2 THE TRAINING DETAILS OF FARE

FARE (Schlarmann et al., 2024) is an unsupervised adversarial fine-tuning scheme designed to obtain a robust CLIP vision encoder, thereby improving robustness across downstream vision tasks. It enforces that the features of adversarially perturbed inputs remain close to those of the unperturbed inputs produced by the original CLIP model. In a class-incremental learning setting, we adapt this

idea by replacing the original CLIP model with the previous CLIP model. Specifically, the loss is defined as:

$$\mathcal{L}_{\text{total}} = \| f_{\theta_t}(\boldsymbol{x}_t^{adv}) - f_{\theta_{t-1}}(\boldsymbol{x}_t) \|_2^2 \tag{27}$$

where $f_{\theta_{t-1}}(\boldsymbol{x}_t)$ denotes the feature representation generated by the model before updating from $\theta_{t-1}$ to $\theta_t$, and $f_{\theta_t}(\boldsymbol{x}_t)$ is the corresponding output after the update.

For S-CIFAR10 and S-STL10, the model is trained for 20 epochs with a learning rate of 0.001, a weight decay of 1e-4, and a batch size of 64. For S-CIFAR100 and S-TinyImageNet, we use the same hyperparameters but extend the training to 50 epochs.

### J.1.3 THE TRAINING DETAILS OF PMG-AFT

PMG-AFT (Wang et al., 2024b) proposes a pretrained model guided adversarial fine-tuning method, which leverages supervision from the original pretrained model through a carefully designed auxiliary branch to enhance robustness. Specifically, it minimizes the distance between the features of adversarial examples in the target model and those in the pretrained model. In a class-incremental learning setting, we adapt this idea by replacing the original CLIP model with the previous CLIP model. The loss is defined as:

$$\mathcal{L}_{\text{total}} = \mathcal{L}_{\text{CE}}(\boldsymbol{q}_t^{adv}, \boldsymbol{y}) + \alpha \cdot \mathcal{L}_{\text{KL}}(\mathcal{S}_t^{adv}|_1^{t-1} \ \| \ \mathcal{S}_{t-1}^{adv}) + \beta \cdot \mathcal{L}_{\text{KL}}(\mathcal{S}_t^{adv} \ \| \ \mathcal{S}_t) \tag{28}$$

The hyperparameters are set to $\alpha = 1.0$ and $\beta = 1.0$. For S-CIFAR10 and S-STL10, the model is trained for 20 epochs with a learning rate of 0.001, a weight decay of 1e-5, and a batch size of 64. For S-CIFAR100 and S-TinyImageNet, we use the same hyperparameters but extend the training to 50 epochs.

### J.1.4 THE TRAINING DETAILS OF TGA-ZSR

TGA-ZSR (Yu et al., 2024b) observes that adversarial perturbations induce a noticeable shift in text-guided attention. To address this, it introduces a simple yet effective strategy that aligns the text-guided attention of adversarial examples obtained from the target model with that of clean examples produced by the original model. In addition, it enforces consistency of text-guided attention between the target and original models on clean examples. In a class-incremental learning setting, we adapt this idea by replacing the original model with the previous model. The loss is defined as:

$$\mathcal{A}(\boldsymbol{x}) = f(\boldsymbol{x}) \cdot g(\boldsymbol{p})^\top, \quad \mathcal{A}(\boldsymbol{x}) \in \mathbb{R}^{P \times 1}$$

$$\mathcal{L}_{\text{total}} = \mathcal{L}_{\text{CE}}(\boldsymbol{q}_t^{adv}, \boldsymbol{y}) + \alpha \cdot \| \mathcal{A}_t(\boldsymbol{x}^{adv})|_1^{t-1} - \mathcal{A}_{t-1}(\boldsymbol{x}) \|_2 + \beta \cdot \| \mathcal{A}_t(\boldsymbol{x})|_1^{t-1} - \mathcal{A}_{t-1}(\boldsymbol{x}) \|_2 \tag{29}$$

We define the text-guided attention as $\mathcal{A}(\boldsymbol{x}) \in \mathbb{R}^{P \times 1}$, where $P$ denotes the number of image patches. The notation $\mathcal{A}_t(\cdot)$ refers to the text-guided attention derived from the current model at task $t$, while $\mathcal{A}_t(\cdot)$ corresponds to that of the previous task model. Moreover, the operator $|_1^{t-1}$ indicates that the attention vector is restricted to the classes observed from tasks 1 through $t-1$.

The hyperparameters are set to $\alpha = 0.08$ and $\beta = 0.05$. For S-CIFAR10 and S-STL10, the model is trained for 20 epochs with a learning rate of 0.001, a weight decay of 1e-5, and a batch size of 64. For S-CIFAR100 and S-TinyImageNet, we use the same hyperparameters but extend the training to 50 epochs.

## J.2 CLASS-INCREMENTAL LEARNING

### J.2.1 THE TRAINING DETAILS OF R-LwF

LwF (Li & Hoiem, 2017) is a regularization-based strategy that mitigates catastrophic forgetting through knowledge distillation. It combines a cross-entropy loss with a distillation loss. To enhance its adversarial robustness, we replace the clean examples in the cross-entropy term with adversarial examples, denoted as R-LwF:

$$\mathcal{L}_{\text{total}} = \mathcal{L}_{\text{CE}}(\boldsymbol{q}_t^{adv}, \boldsymbol{y}) + \alpha \cdot \mathcal{L}_{\text{KL}}(\mathcal{S}_t|_1^{t-1} \ \| \ \mathcal{S}_{t-1}) \tag{30}$$

The hyperparameter is set to $\alpha = 1.0$. For S-CIFAR10 and S-STL10, the model is trained for 20 epochs with a learning rate of 0.1, a weight decay of 1e-5, and a batch size of 64. For S-CIFAR100 and S-TinyImageNet, the model is trained for 50 epochs with a learning rate of 0.5, a weight decay of 1e-5, and a batch size of 64.

### J.2.2 THE TRAINING DETAILS OF R-LwF-MC

LwF-MC (Dhar et al., 2019) is an improved variant of LwF that replaces both the cross-entropy loss and the KL-divergence term with a binary cross-entropy (BCE) loss, allowing each output dimension to be treated independently and thereby alleviating catastrophic forgetting more effectively. To enhance its adversarial robustness, we replace the clean examples in the first BCE term with adversarial examples, denoted as R-LwF-MC:

$$ratio = len(\boldsymbol{y}_{t-1}) \ / \ len(\boldsymbol{y}_t)$$

$$\mathcal{L}_{\text{total}} = (1 - ratio) \cdot \mathcal{L}_{\text{BCE}}(\mathcal{S}_t^{adv}|_{t-1}^t, \mathbf{1}_{\boldsymbol{y}}) + ratio \cdot \mathcal{L}_{\text{BCE}}(\mathcal{S}_t|_1^{t-1}, sigmoid(\mathcal{S}_{-1})) \tag{31}$$

which $\mathbf{1}_{\boldsymbol{y}}$ denotes the binary one-hot vector indicating the ground-truth class $\boldsymbol{y}$, $sigmoid(\cdot)$ denotes the sigmoid activation function, and $ratio$ is defined as the proportion of previously learned relative to the total number of classes after task $t$.

For S-CIFAR10 and S-STL10, the model is trained for 20 epochs with a learning rate of 0.1, a weight decay of 1e-5, and a batch size of 64. For S-CIFAR100 and S-TinyImageNet, the model is trained for 50 epochs with a learning rate of 0.5, a weight decay of 1e-5, and a batch size of 64.

### J.2.3 THE TRAINING DETAILS OF R-EWC-ON

EWC (Kirkpatrick et al., 2017) is are regularization-based approach that constrains parameter updates based on their estimated importance to previously learned tasks. By discouraging significant changes to critical parameters, EWC effectively preserves prior knowledge and mitigates catastrophic forgetting. To enhance its adversarial robustness, we replace the clean examples in the cross-entropy term with adversarial examples, denoted as R-EWC-on:

$$\mathcal{L}_{\text{total}} = \mathcal{L}_{\text{CE}}(\boldsymbol{q}_t^{adv}, \boldsymbol{y}) + \sum_i \frac{\alpha}{2} F_i(\theta_t^i - \theta_{t-1}^i)^2 \tag{32}$$

where $F_i$ is the Fisher information estimating the importance of parameter $\theta^i$, $\theta_t$ denotes the model parameters after learning task $t$, and $\alpha$ controls the strength of the regularization term.

The hyperparameter $\lambda$ is set to 25. The model is trained with a weight decay of 1e-5 and a batch size of 32. For S-CIFAR10 and S-STL10, the model is trained for 20 epochs with a learning rate of 0.1, while for S-CIFAR100 and S-TinyImageNet, it is trained for 50 epochs with a learning rate of 0.5.

### J.2.4 THE TRAINING DETAILS OF R-SI

SI (Zenke et al., 2017) builds on a concept similar to EWC but introduces intelligent synapses that incorporate aspects of biological plasticity into artificial neural networks. Each synapse incrementally accumulates task-relevant information and uses this knowledge to efficiently integrate new memories while preserving previously acquired ones, thereby mitigating catastrophic forgetting. TO enhance its adversarial examples, we replace the clean examples in the cross-entropy term with adversarial examples, denoted as R-SI:

$$\mathcal{L}_{\text{total}} = \mathcal{L}_{\text{CE}}(\boldsymbol{q}_t^{adv}, \boldsymbol{y}) + \alpha \cdot \sum_i \Omega_i(\theta_t^i - \theta_{t-1}^i)^2 \tag{33}$$

where $\Omega_i$ is each synapse incrementally accumulates task-relevant information, and $c$ controls the strength of the regularization term.

The hyperparameter $\alpha$ is fixed at 0.5. The model is trained with a weight decay of 1e-5 and a batch size of 64. For S-CIFAR10 and S-STL10, the model is trained for 20 epochs with a learning rate of 0.1, while for S-CIFAR100 and S-TinyImageNet, it is trained for 50 epochs with a learning rate of 0.5.

## J.3 CLASS-INCREMENTAL LEARNING WITH CLIP

### J.3.1 THE TRAINING DETAILS OF R-PROOF

Proof (Zhou et al., 2025) trains task-specific projection layers on top of frozen image and text encoders. For each task, additional projections are introduced while the previous ones remain fixed,

thereby mitigating the forgetting of previously learned concepts. Furthermore, a fusion module is incorporated to better exploit cross-model information. By jointly refining visual and textual representations, the model captures richer task-specific semantic information that facilitates recognition. To enhance its adversarial robustness, we replace the clean examples with adversarial examples, leading to our variant termed R-Proof:

$$PI_t(\boldsymbol{x}^{adv}) = \sum_{m=1}^{b} PI_t^m(f(\boldsymbol{x}^{adv})), \quad PT_t(\boldsymbol{p}) = \sum_{m=1}^{b} PT_t^m(g(\boldsymbol{p}))$$

$$\mathbf{P}_t = [PI_t(pro_1), PI_t(pro_2), \quad PI_t(pro_b)], \quad \mathbf{T}_t = [PT_t(\boldsymbol{p}_1), PT_t(\boldsymbol{p}_2), \cdots, PT_t(\boldsymbol{p}_b)]$$

$$\mathbf{C} = [\boldsymbol{c}_1, \boldsymbol{c}_2, \cdots, \boldsymbol{c}_b], \quad [\widetilde{PI}_t(\boldsymbol{x}^{adv}), \tilde{\mathbf{P}}_t, \tilde{\mathbf{T}}_t, \tilde{\mathbf{C}}] = Attn([PI_t(\boldsymbol{x}^{adv}), \mathbf{P}_t, \mathbf{T}_t, \mathbf{C}])$$

$$S_{i,j}^{pm} = \tau \cdot \frac{PI_t(\boldsymbol{x}_i^{adv})}{\left\| PI_t(\boldsymbol{x}_i^{adv}) \right\|} \cdot \left( \frac{PT_t(\boldsymbol{p}_j)}{\left\| PT_t(\boldsymbol{p}_j) \right\|} \right)^\top, \quad \boldsymbol{q}^{pm} = \frac{S_{i,j}^{pm}}{\sum_k S_{i,k}^{pm}}$$

$$S_{i,j}^{vm} = \tau \cdot \frac{\widetilde{PI}_t(\boldsymbol{x}_i^{adv})}{\left\| \widetilde{PI}_t(\boldsymbol{x}_i^{adv}) \right\|} \cdot \left( \frac{\widetilde{PI}_t(pro_j)}{\left\| \widetilde{PI}_t(pro_j) \right\|} \right)^\top, \quad \boldsymbol{q}^{vm} = \frac{S_{i,j}^{vm}}{\sum_k S_{i,k}^{vm}} \tag{34}$$

$$S_{i,j}^{tm} = \tau \cdot \frac{\widetilde{PI}_t(\boldsymbol{x}_i^{adv})}{\left\| \widetilde{PI}_t(\boldsymbol{x}_i^{adv}) \right\|} \cdot \left( \frac{\widetilde{PT}_t(\boldsymbol{p}_j)}{\left\| \widetilde{PT}_t(\boldsymbol{p}_j) \right\|} \right)^\top, \quad \boldsymbol{q}^{vm} = \frac{S_{i,j}^{tm}}{\sum_k S_{i,k}^{tm}}$$

$$\mathcal{L}_{\text{total}} = \mathcal{L}_{\text{CE}}(\boldsymbol{q}^{pm}, \boldsymbol{y}) + \mathcal{L}_{\text{CE}}(\boldsymbol{q}^{vm}, \boldsymbol{y}) + \mathcal{L}_{\text{CE}}(\boldsymbol{q}^{tm}, \boldsymbol{y})$$

Here, $PI_t^m(\boldsymbol{x}^{adv})$ and $PT_t^m(\cdot)$ denote the $m$-th image and text projections, respectively, and their sum yield $PI_t(\boldsymbol{x}^{adv})$ and $PT_t(\boldsymbol{p})$. The collections of projected features are denoted as $PI_t(\boldsymbol{x}^{adv}$ for adversarial images, $\mathbf{P}_t$ for visual prototypes, $\mathbf{T}_t$ for text prompts, and $\mathbf{C}$ for additional context embeddings.These features are then refined by the attention module, producing refined representations $\widetilde{PI}_t(\boldsymbol{x}^{adv})$, $\tilde{\mathbf{P}}_t$, $\tilde{\mathbf{T}}_t$, and $\tilde{\mathbf{C}}$. Based on these fine features, three types of similarity scores are computed: $S_{i,j}^{pm}$ for projected matching between adversarial examples and text prompts, $S_{i,j}^{vm}$ for visual matching between adversarial examples and visual prototypes. and $S_{i,j}^{tm}$ for text matching between adversarial examples and text prompts. Each similarity score is normalized across all candidate classes, resulting in probability distributions $\boldsymbol{q}^{pm}$, $\boldsymbol{q}^{vm}$, and $\boldsymbol{q}^{tm}$, respectively. These distributions are then used to compute the total loss, which combines cross-entropy terms over three matching perspectives.

The model is trained with a learning rate of 0.001, a weight decay of 0.05, and a batch size of 64. For S-CIFAR10 and S-STL10, the model is trained for 20 epochs, while for S-CIFAR100 and S-TinyImageNet, it is trained for 50 epochs.

### J.3.2 THE TRAINING DETAILS OF R-RAPF

RAPF (Huang et al., 2024) introduces a linear adapter layer $W$ appended to the image encoder. After fine-tuning this adapter, it employs a decomposed parameter fusion method to integrate parameters from both the new and old adapters. To enhance category separation, RAPF computes distances between new and old text embeddings and leverages these distances to select statistical features of hard examples from previous tasks for sampling, thereby guiding the fine-tuning of the adapter on new tasks. To enhance its adversarial robustness, we replace the clean examples in the cross-entropy term with adversarial examples and compute statistical features from these adversarial examples for sampling, denoted as R-RAPF:

$$\mathcal{L}_{\text{total}} = \mathcal{L}_{\text{CE}}(\boldsymbol{q}_t^{adv}, \boldsymbol{y}) + \mathcal{L}_{\text{hinge}}$$

$$\mathcal{L}_{\text{hinge}} = \sum_{k=1}^{|\mathcal{P}|} \max(dist(W(\boldsymbol{e}^{adv}), g(\boldsymbol{p}_t)) - dist(W(\boldsymbol{e}^{adv}), g(\boldsymbol{p}_{1:t-1})) + m, 0) \tag{35}$$

where $\mathcal{P} = \{(i,j)|d_{i,j} < \lambda\}$, $d_{i,j} = dist(g(\boldsymbol{p}_t), g(\boldsymbol{p}_{1:t-1}))$, $dist(\cdot, \cdot)$ denotes the Euclidean distance, and $m$ is a constant margin. The sampled data of the old category $\boldsymbol{q}_{\text{old}}$, derived from the statistical features of adversarial examples, is denoted by $\boldsymbol{e}^{adv}$. In R-RAPF, $\mathcal{L}_{\text{CE}}$ is used to update all parameters of the model, including both the image encoder and the adapter, whereas $\mathcal{L}_{\text{hinge}}$ is applied exclusively to update the adapter.

We employ the text prompt template "a good photo of a {}". The hyperparameters are set as $\lambda = 0.5$, $m = 1.0$ and the max ratio to 0.6. The model is trained with a learning rate of 0.01, a weight decay of 0, and a batch size of 64. For S-CIFAR10 and S-STL10, the model is trained for 20 epochs, while for S-CIFAR100 and S-TinyImageNet, it is trained for 50 epochs.

### J.3.3 THE TRAINING DETAILS OF R-SG

SG (Yu et al., 2024a) leverages semantic information as auxiliary knowledge to improve the effectiveness of class-incremental learning. Specifically, it utilizes intra-task semantic relationships to generate more informative labels for the current task. Furthermore, it exploits inter-task semantic relationships to enhance knowledge distillation, thereby mitigating the forgetting of previously acquired knowledge. To enhance its adversarial robustness, we replace the clean examples with adversarial examples, denoted as R-SG:

$$\mathcal{L}_{\text{total}} = \mathcal{L}_{\text{CE}}(\boldsymbol{q}_t^{adv}, \boldsymbol{y}) + \alpha \cdot \mathcal{L}_{\text{KL}}(\boldsymbol{q}_t^{adv} \ || \ \boldsymbol{y}^{sg}) + \beta \cdot \mathcal{L}_{\text{KL}}(\boldsymbol{q}_t^{adv}|_1^{t-1} \ || \ \boldsymbol{q}_{t-1}^{adv})$$

$$\boldsymbol{y}_{i,j}^{sg} = \frac{\mathcal{S}_{i,j}^{c \leftrightarrow c}}{\sum_k \mathcal{S}_{i,k}^{c \leftrightarrow c}}, \quad \mathcal{S}_{i,j}^{c \leftrightarrow c} = \tau \cdot \frac{g(\boldsymbol{p}_i)}{\|g(\boldsymbol{p}_i)\|} \cdot (\frac{g(\boldsymbol{p}_j)}{\|g(\boldsymbol{p}_j)\|})^{\top} \tag{36}$$

Here, $\boldsymbol{y}_{i,j}^{sg}$ represents semantically-guided labels. $\boldsymbol{q}_t^{adv}|_1^{t-1}$ refers to the predictions of the new-task model restricted to the classes from previously learned tasks, while $\boldsymbol{q}_{t-1}^{adv}$ represents the predictions by the model trained on those previous tasks.

We employ the text prompt template "a bad photo of {}". The hyperparameters are set as $\alpha = 0.5$ and $\beta = 0.5$. For S-CIFAR10, the model is trained for 20 epochs with a learning rate of 0.1, a weight decay of 2e-4, and a batch size of 64, whereas for S-STL10, the learning rate is set to 0.01 with the same weight decay and batch size. For S-CIFAR100 and S-TinyImageNet, the model is trained for 50 epochs with a learning rate of 0.1, a weight decay of 2e-4, and a batch size of 64.

### J.4 ROBUST CLASS-INCREMENTAL LEARNING

### J.4.1 THE TRAINING DETAILS OF TABA

TABA (Bai et al., 2023) adopts an idea similar to LwF-MC, but differs in that it enhances data diversity through augmentation. Specifically, it first selects boundary samples, as these samples are more vulnerable to attacks, and denotes this set as $\mathcal{B}$. Then, it applies mixup between the current task's data set $\mathcal{B}_t$ and the previous tasks' data set $\mathcal{B}_o$, and defines the resulting set as $\mathcal{D}_{\text{TABA}}$. Finally, training is performed jointly on the current task data set $\mathcal{D}_t$ and $\mathcal{D}_{\text{TABA}}$.

$$ratio = len(\boldsymbol{y}_{t-1}) \ / \ len(\boldsymbol{y}_t)$$

$$\mathcal{L}_{\text{total}} = (1 - ratio) \cdot \mathcal{L}_{\text{BCE}}(\mathcal{S}_t^{adv}|_{t-1}^t, \mathbf{1}_{\boldsymbol{y}}) + ratio \cdot \mathcal{L}_{\text{BCE}}(\mathcal{S}_t^{adv}|_1^{t-1}, sigmoid(\mathcal{S}_{t-1})) \tag{37}$$

TABA is evaluated under the replay-based setting. The model is trained with a weight decay of 1e-5 and a batch size of 64. For S-CIFAR10 and S-STL10, we use a learning rate of 0.1 and train for 20 epochs, while for S-CIFAR100 and S-TinyImageNet, we use a learning rate of 0.5 and train for 50 epochs.

### J.4.2 THE TRAINING DETAILS OF FLAIR

FLAIR (Cho et al., 2025) systematically establishes a framework for robust class-incremental learning. It first explores a series of baselines that combine incremental learning with existing adversarial training methods and observes that such integration leads to conflicts between acquiring new knowledge and retaining previously learned knowledge. It then further investigates this challenge by analyzing the output differences between clean and adversarial examples through a Taylor expansion, revealing that these discrepancies are governed by the model's gradients and Hessians. It can be math:

$$\mathcal{L}_{total} = \mathcal{L}_{BCE}(\mathcal{S}_t^{adv}|_{t-1}^t, \mathbf{1}_y) + \alpha \cdot \mathcal{L}_{\text{BCE}}(\mathcal{S}_t^{adv}|_1^{t-1}, sigmoid(\mathcal{S}_{t-1}^{adv}))$$

$$+ \beta \cdot \mathcal{L}_{KL}(\mathcal{S}_t^{adv}|_1^{t-1} - \mathcal{S}_t|_1^{t-1} \ || \ \mathcal{S}_{t-1}^{adv} - \mathcal{S}_{t-1}) \tag{38}$$

where $sigmoid(\cdot)$ denotes the sigmoid activate function.

The hyperparameters are set as $\alpha = 0.5$ and $\beta = 2.0$. For S-CIFAR10 and S-STL10, the model is trained for 20 epochs with a learning rate of 0.1, a weight decay of 1e-5, and a batch size of 64. For S-CIFAR100, the model is trained for 50 epochs with a learning rate of 0.5, a weight decay of 1e-5, and a batch size of 64, whereas for S-TinyImageNet, the learning rate is set to 1.0 with the same weight decay and batch size.

