# OpenReview forum: "From Forgetting to Robustness: Robust Class-Incremental Learning with CLIP"
_ICLR.cc/2026/Conference — Submitted to ICLR 2026_

### Official Review · Reviewer_j5be · 2025-10-16

**Soundness:** 2
**Presentation:** 2
**Contribution:** 4
**Rating:** 6
**Confidence:** 5

**Summary:**

This paper reinterprets the concept of combining Adversarial Training (AT) and Class-Incremental Learning (CIL) by leveraging the CLIP framework, extending ideas originally explored in TABA and FLAIR. While the prior works approached AT + CIL through conventional vision backbones (like ResNet), this paper introduces the notion of RCIL (Robust Class-Incremental Learning)—performing AT + CIL directly within CLIP. The proposed method, SAGE (Selective parameter optimization for Adversarial training with GEometric constraint), selectively updates only the most critical parameters during adversarial learning while enforcing a geometric constraint to align clean and adversarial embeddings. By applying the CLIP-based contrastive paradigm to incremental and adversarial settings, SAGE bridges robustness and continual learning more effectively than earlier AT + CIL frameworks, yielding a more practical solution for AT + CIL.

After reviewing the main paper, supplementary material, and attached code, I find the work to be generally well-supported and leaning toward acceptance; however, several weaknesses and open questions remain, which I outline below.

**Strengths:**

**1. Well-structured and comprehensive framework**

The proposed framework — AT + CIL on CLIP (referred to as RCIL) — is well constructed, with a solid experimental setup and clearly chosen baselines. The authors include reasonable comparative methods such as AT fine-tuning approaches on CLIP (e.g., TeCoA, FARE), and CLIP-based CIL variants (e.g., R-SG). Moreover, previous AT + CIL methods such as FLAIR are systematically reinterpreted within a unified RCIL framework, extending their original residual-family formulation to a CLIP-based setting. Overall, the experiments are formalized and carefully structured, providing a consistent and standardized evaluation of the proposed framework.

**2. Simple yet effective and computationally efficient design**

The proposed method is simple yet effective, offering a practical solution to the high computational burden typically associated with AT. By selectively updating only the most important parameters, SAGE reduces unnecessary computation while maintaining both robustness from forgetting and robust learnability in the RCIL setting.

**3. Reproducibility and implementation clarity**

The paper provides clear implementation details and includes well-structured pseudo-code in Algorithm 1, along with the released code in the supplementary materials. After examining both the text and the attached code, I find that the overall training pipeline is logically organized and reproducible in principle. (Note: I did not directly execute the provided code to verify the numerical results in the tables.)

Considering these three strengths, the paper establishes a solid and credible foundation for future research on AT + CIL. In this sense, it could serve as an important baseline and reference point for subsequent studies.

**Weaknesses:**

## Major Weaknesses
**1. Conceptual limitations in the Observation section**
While the Observation section (Sec. 4.1) offers an intuitive theoretical framing, its formulation in Eq. (8) oversimplifies the notion of robustness. The objective assumes that minimizing the input-gradient term
$
\|\nabla_x f_{\theta_t}(x_t)\|^2
$
leads to improved robustness. However, this assumption only holds if the model already produces correct predictions on clean samples at task $t$. If the model fails to learn the clean data properly, reducing its input gradients merely enforces local flatness around an incorrect decision region, offering no true robustness improvement. More importantly, Eq. (8) is introduced as a conceptual formulation but is never explicitly optimized or referenced later in the method.


**2. Potential implementation inconsistency in Table 2 results**

Table 2 reports nearly zero clean accuracy and zero BWT for methods such as R-LwF-MC and FLAIR on S-CIFAR100 and S-TinyImageNet. A BWT of 0 combined with almost 0 final accuracy indicates that the model likely failed to learn the first task at all, rather than preserving previous knowledge. The paper attributes this to the use of BCE loss instead of CE, but this explanation appears insufficient. In the original FLAIR paper, experiments under ResNet-18 did not exhibit such collapse.
It appears that the unexpected behavior might be related to how BCE is implemented in the attached code. BCE is applied after feature normalization within the embedding space, which could potentially reduce the logit magnitude and make optimization unstable. While this is only a conjecture, clarifying whether feature normalization is related to the BCE computation would help ensure interpretability of the reported results.

## Minor Weaknesses
The following points are relatively minor and do not significantly affect my overall evaluation or score. If the relevant experimental data are not already available, there is no need to conduct additional experiments.

**1. Adversarial strength selection in training and evaluation**
The choice of $\epsilon = 1/255$ for adversarial training appears rather weak compared to common settings in similar CLIP-based robust learning studies (e.g., FARE uses $\epsilon = 4/255$). In Table 3, it is also unclear whether the results averaged across $\epsilon = \{1/255, 2/255, 4/255\}$ are all obtained from models trained with $\epsilon = 1/255$, or if separate models were trained for each setting. Training and evaluating models under matched $\epsilon$ values (e.g., $1/255$–$1/255$, $2/255$–$2/255$, $4/255$–$4/255$) and reporting each table separately would make the robustness analysis clearer and more interpretable.

**2. Backbone limitation**
All experiments are conducted solely on the ViT-B/32 backbone. It would be informative to evaluate whether the proposed SAGE framework maintains similar robustness-forgetting trade-offs on larger or stronger backbones such as ViT-L/14, as done in prior work like FARE in the context of CLIP robust finetuning.

**3. Numerous typos and inconsistent terminology**
Examples include:
- “noes” → should be “ones” (Abstract, 012)
- “FLIAR” → inconsistent with the main text’s “FLAIR” (Related Work, 064)
- “text prompt prompt” → redundant phrase (Training details, 359)
- “Similarity, the vector b ...” → should be “Similarly, the vector b ...” (Appendix C, 754)
- “derived form” → should be “derived from” (Appendix H.1.4, 966)
- “adversarail” → should be “adversarial” (Appendix H.3.2, 1082)
- “us applied” → should be “is applied” (Appendix H.3.2, 1092)
- “sigmod” → should be “sigmoid” (Appendix H.4.2, 1143)
- Notation inconsistencies in equations, such as the inconsistent use of BCE in Eq. (39)

**Questions:**

1. Is there a specific reason for using 2 iterations with a step size of 1/255 (equal to the attack budget) for training attacks, and 10 iterations with the same step size for test attacks? This setup seems somewhat different from the conventional AT practice, where 10–20 PGD steps with a smaller step size than the attack budget are typically used.

2. In the original FLAIR paper, data augmentation was reported to improve performance (FLAIR+). Does this observation also hold for SAGE, or have you noticed a similar benefit when applying data augmentation?

3. Robust fine-tuning methods for CLIP like FARE exhibit a forgetting of “natural” features. Since SAGE also fine-tunes a pre-trained CLIP, could you comment on whether similar forgetting occurs at the first task? In particular, how much does the zero-shot CIFAR-10 clean accuracy of the pre-trained CLIP drop after training only the first task with SAGE?

4. If SAGE were initialized not from vanilla CLIP but from a model that has already been robustly fine-tuned on ImageNet (via FARE), would you expect further gains?

5. Since SAGE is built upon CLIP, have you explored or considered its zero-shot accuracy or robustness, similar to prior CLIP-based robust finetuning like FARE?

6. In Equation (28) in Section H.1.2, it seems that the loss function of FARE may be incorrectly formulated. Shouldn't the adversarial input be fed into the current task model?

7. “This is a photo of {}” is the commonly used CLIP prompt as noted in L359. However, you set the prompt to "a good photo of a {}" at L1093 and to "a bad photo of {}" at L1114. Is there a specific reason to prefer the “good/bad photo” templates in these sections? Have you checked how sensitive the results are to the prompt template—e.g., replacing them with the standard "This is a photo of {}"?


Overall, I am leaning toward accept, as the paper proposes a clear and meaningful contribution.
However, several technical and clarity concerns remain.
If these issues are adequately addressed during the rebuttal, it may lead me to adjust my score accordingly.

---

> ### Author Response · Authors · 2025-11-20
>
> **Q1: The observation section.**
>
> **A1:** The reviewer is correct that improving adversarial robustness requires not only reducing the input-gradient term, as discussed in Eq. (6), but also ensuring that the model maintains high accuracy on clean examples. The main purpose of the observation section is not to present a single new equation, but to provide insight into the interaction between adversarial robustness and class-incremental learning. The claim that the two terms are 'contradictory' is primarily an insight from our analysis of their objectives, supported by empirical evidence. Adversarial training encourages feature representations to be locally invariant around each input, effectively 'flattening' the feature space for robustness. In contrast, class-incremental learning objectives aim to preserve class boundaries and maintain discriminative features from previous tasks. These objectives can pull feature representations in opposing directions, creating a trade-off that has not been explicitly discussed in prior work. We will also consider deleting or revising Eq. (8) to avoid confusion.
>
> This theoretical observation is further validated empirically in **Appendix B**, where naive combinations of adversarial training and standard CIL significantly increase forgetting or reduce robustness. Thus, the 'contradiction' reflects both the conflicting nature of the optimization objectives and the observed empirical behavior, motivating the need for methods like SAGE to balance these objectives effectively.
>
>
> **Q2: The BCE with CLIP.**
>
> **A2:** To understand why FLAIR and R-LwF-MC behaved abnormally only in the CLIP-based continual learning setting, we analyzed the loss functions they rely on. Both FLAIR and R-LwF-MC adopt Binary Cross-Entropy (BCE) rather than Cross-Entropy (CE). This choice is effective for standard DNNs, but we speculate that it may be less suitable for CLIP, where text embeddings across classes tend to be highly correlated. The independence assumption of BCE may therefore be weakened, and the lack of normalization and inter-class competition could make it more challending to separate semantically similar classes. Prior work[1] has also reported this limitation.
>
> To further verify that BCE is the source of the problem, we replaced the BCE loss in R-LwF-MC and FLAIR with CE loss while keeping all other implementations unchanged. The experimental results in Table S7 show that replacing BCE with CE eliminates the issues observed previously, leading to more stable and accurate predictions. We will provide additional experiments on this in the revised version.
>
> Table S7: Experiments on S-CIFAR100 replacing the BCE loss with CE loss in R-LwF-MC and FLAIR.
> |Method |Clean $\overline{A}$ |Clean $A_{last}$ |Clean $BWT$  |PGD $\overline{A}$   |PGD $A_{last}$   |PGD $BWT$    |Auto. $A_{last}$|
> |------|------|------|------|------|------|------|------|
> |R-LwF-MC   |28.24  |9.66   |-92.22 |24.82  |8.88   |-82.73 |8.85|
> |FLAIR  |28.57  |9.52   |-91.90 |24.88  |8.92   |-82.26 |8.91|
> |SAGE   |**63.20**  |**49.02**  |**-22.62** |**48.49**  |**35.59**  |**-19.89** |**28.98**|
>
> Regarding the reviewer's question about whether feature normalization interacts with BCE computation, we performed a direct test. We removed feature normalization and computed BCE on the raw features. The optimization immediately became unstable and the loss produced abnormal values that stopped training. This shows that the issue does not come from normalization itself.
>
> [1]Li Q, Xiao H, Shen L. BCE vs. CE in Deep Feature Learning[C]//Forty-second International Conference on Machine Learning.

---

> ### Author Response · Authors · 2025-11-20
>
> **Q3: Impact of adversarial attack strength.**
>
> **A3:** We train our models under a relatively weak attack strength of 1/255 and then evaluate them at 1/255, 2/255, and 4/255, corresponding to the settings you mentioned (1/255–1/255, 1/255–2/255, 1/255–4/255). This approach allows us to assess the model’s robustness when trained on weak adversarial examples and tested against stronger attacks, reflecting performance under more diverse conditions. To make the robustness analysis clearer and more interpretable, we report the results for each attack strength separately, as shown in Table S10. The experiments show that SAGE achieves the highest average robustness. Overall, it consistently outperforms other methods under commonly used attack conditions, demonstrating its effectiveness in balancing robustness across varying attack strengths.
>
> Table S10: Evaluation of several methods on ViT-B/32 without memory. We report PGD-10 $A_{last}$ on S-CIFAR10 under attack strengths of 1/255, 2/255, and 4/255.
> |Type | Method | 1/255 | 2/255 | 4/255 | Average |
> |------|------|------|------|------|------|
> | AT           | TeCoA       |9.20 |3.80 |0.17 |4.39 |
> |              | FARE        |27.33 |2.59 |0.21 |10.04 |
> |              | PMG-AFT      |9.11 |4.18 |0.54 |4.61 |
> |              | TGA-ZSR      |15.70 |8.78 |5.65 |10.04 |
> | R-CIL        | R-LwF         |19.38 |19.05 |**19.01** |19.15 |
> |              | R-LwF-MC     |24.08 |20.29 |17.33 |20.57 |
> |              | R-EWC-on      |19.38 |18.80 |18.69 |18.96 |
> |              | R-SI         |19.38 |18.90 |18.86 |19.05 |
> | R-CIL-CLIP  | R-RAPF       |19.34 |18.24 |17.80 |18.46 |
> |              | R-SG        |18.46 |13.63 |11.17 |14.42 |
> |              | R-Proof       |13.61 |12.96 |11.73|12.77 |
> | RCIL         | FLAIR        |32.31 |24.30 |17.92 |24.84 |
> | RCIL4CLIP    | SAGE        |**47.85** |**30.89** |13.00 |**30.24** |
>
> **Q4: Larger backbone (ViT-L/14).**
>
> **A4:** To further examine whether our method scales to large-scale CLIP variants, we extend the evaluation to ViT-L/14. The final results in Table S4 show that on the larger and stronger ViT-L/14 most methods experience even more severe forgetting under adversarial conditions. This is likely because full parameter tuning makes it harder for the model to retain previous tasks. SAGE still maintains a clear and stable advantage by improving both clean accuracy and adversarial robustness. The overall trend shows that the robustness and scalability of SAGE continue to hold when applied to larger backbones.
>
> Table S4: Evaluation of several methods on ViT-L/14 without memory. We report Clean, PGD-10 accuracy (%), and BWT on S-CIFAR10 under an attack strength of 1/255.
>
> |Type   |Method |Clean $\overline{A}$ |Clean $A_{last}$ | Clean $BWT$ | PGD $\overline{A}$  | PGD $A_{last}$ | PGD $BWT$|
> |------|------|-------|------|------|------|------|------|
> |AT |TeCoA  |45.27  |19.81  |-98.99 |43.88  |19.56  |-95.40|
> |    |FARE  |50.63  |13.85  |**-50.53** |42.26  |12.87  |**-47.75**|
> |R-CIL  |R-LwF-MC   |44.46  |20.20  |-93.38 |41.04  |19.22  |-89.83|
> |    |R-EWC-on  |39.15  |17.92  |-84.36 |37.05  |16.93  |-82.14|
> |    |R-SI  |44.39  |19.52  |-96.80 |42.86  |18.99  |-48.83|
> |R-CIL-CLIP |R-Proof    |29.90  |10.00  |-61.24 |28.44  |10.00  |-58.26|
> |RCIL   |FLAIR  |43.71  |19.72  |-95.08 |41.69  |19.30  |-91.24|
> |RCIL4CLIP  |SAGE   |**61.93**  |**34.19**  |-77.55 |**56.59**  |**28.36**  |-78.86|
>
> **Q5: Typos and Equations.**
>
> **A5:** We have thoroughly reviewed the manuscript for minor errors and terminology inconsistencies, making all necessary corrections in the revised version. Concerning the comment on notation in Eq. (39), we have verified the equations and confirm that the use of BCE is consistent throughout the FLAIR[1] manuscript.
>
> [1]Cho S, Lee H, Kim C. Enhancing Robustness in Incremental Learning with Adversarial Training[C]//Proceedings of the AAAI Conference on Artificial Intelligence. 2025, 39(3): 2518-2526.

---

> ### Author Response · Authors · 2025-11-20
>
> **Q6: Experiments with a smaller step size.**
>
> **A6:** In FLAIR [1], the same step size (2/255) is used for both training and testing. Accordingly, we follow FLAIR and maintain the same step size during both training and testing. To align with standard adversarial training practice, where 10–20 PGD steps with a step size smaller than the attack budget are typically used, we reduce the test-time step size from 1/255 by a factor of 10, corresponding to the number of iterations, when evaluating on S-CIFAR10, as reported in Table S11. Interestingly, under this reduced step size, our method exhibits a performance gain, whereas FLAIR experiences a notable drop in robustness, further highlighting the stability of our approach under stronger evaluation conditions.
>
> Table S11: Experiments on S-CIFAR10 with a smaller step size under an attack strength of 1/255.
> | Step Size | Method | PGD $\overline{A}$ | PGD $A_{last}$ | BWT_adv |
> |-----------|---------|-------|----------|------|
> | 1/255     | FLAIR  | 51.73 | 32.31 | -77.91   |
> |           | SAGE   | **61.59** | **46.83** | **-43.82**   |
> | 0.1/255   | FLAIR  | 53.68 |29.81 | -83.06   |
> |           | SAGE   | **64.87** |**51.58** | **-41.61**   |
>
> [1]Cho S, Lee H, Kim C. Enhancing Robustness in Incremental Learning with Adversarial Training[C]//Proceedings of the AAAI Conference on Artificial Intelligence. 2025, 39(3): 2518-2526.
>
> **Q7: Experiments with data augmentation.**
>
> **A7:** To examine whether augmentation benefits our method, we additionally follow FLAIR and apply RandAugment (RA) on S-CIFAR10. The results in Table S12 show that RA provides improvements across both clean and adversarial metrics.
>
> Table S12: Experiments on S-CIFAR10 with data augmentation under an attack strength of 1/255.
> |Method	|Clean $\overline{A}$| Clean	$A_{last}$|Clean 	$BWT$	| PGD $\overline{A}$| PGD	$A_{last}$| PGD	$BWT$ |Auto.	$A_{last}$ |
> |-------|------|-------|------|------|------|------|------|
> |SAGE|	72.36	|63.67|	**-34.12**|	61.75|	47.85|	**-42.27**|	41.60|
> |SAGE+|	**72.53**|	**64.98**|	-34.58|	**62.98**|	**50.18**|	-42.38|	**44.74**|
>
>
> **Q8: Impact of SAGE on CLIP's zero-shot accuracy.**
>
> **A8:** We evaluated zero-shot performance after training each of the five tasks on S-CIFAR10 and verified on Tiny-ImageNet. The original CLIP achieves 57.26% zero-shot clean accuracy and 0.88% zero-shot adversarial accuracy on Tiny-ImageNet. As shown in Table S13, after fine-tuning, zero-shot performance decreases for all methods as tasks progress. Nevertheless, SAGE remains competitive, falling slightly below only those methods explicitly designed for zero-shot adversarial robustness.
>
> Table S13: Zero-shot adversarial robustness and clean accuracy after training.
> |	|TeCoA | FARE | PMG-AFT | TGA-ZSR || R-LwF | R-LwF-MC | R-EWC-on | R-SI | R-RAPF | R-SG | R-Proof || FLAIR | SAGE|
> |------|------|------|------|------|------|------|------|------|------|------|------|------|------|------|------|
> |Robust (%) |0.00|2.71|0.00|0.64||0.85|0.82|0.85|0.81|0.84|0.51|0.39||0.91|**2.75**|
> |Clean (%) |0.73|**35.79**|0.51|12.38||0.94|1.90|0.91|0.92|0.59|0.48|0.44||2.39|8.25|

---

> ### Author Response · Authors · 2025-11-20
>
> **Q9: Evaluating performance with robust backbone initialization.**
>
> **A9:** We initialized the model using a FARE [1] pretrained Tiny-ImageNet checkpoint. As shown in Table S14, this initialization leads to consistent improvements on S-CIFAR10, S-STL10, and S-TinyImageNet, while yielding a small drop on S-CIFAR100. The overall trend shows that robust initialization helps SAGE adapt more confidently to most continual learning scenarios and encourages a more stable progression of robustness. We have reviewed the manuscript, and the experiment is provided in **Appendix I**.
>
> Table S14: Evaluation of several methods on ViT-B/32 without memory using robust backbone initialization. We report average PGD-10, Auto. accuracy (%) and BWT on S-CIFAR10, S-STL10, S-CIFAR100 and S-TinyImageNet under an attack strength of 1/255.
>
> |Datasets   |Methods    |Clean $\overline{A}$ |Clean $A_{last}$ |Clean $BWT$  |PGD $\overline{A}$   |PGD $A_{last}$   |PGD $BWT$    |Auto. $A_{last}$    |
> |------|------|------|------|------|------|------|------|------|
> |S-CIFAR10  |FLAIR  |61.27  |45.83  |-66.05 |51.73  |32.31  |-77.91 |30.90|
> |   |FLAIR+FARE |48.15  |25.66  |-91.76 |44.34  |20.26  |-94.80 |19.51|
> |    |SAGE  |72.36  |63.67  |**-34.12** |61.75  |47.85  |**-42.27** |41.60|
> |    |SAGE+FARE |**73.52**  |**66.86**  |-34.24 |**62.22**  |**50.33**  |-45.00 |**45.52**|
> |S-STL10    |FLAIR  |71.32  |64.92  |-42.17 |59.69  |48.26  |-51.59 |41.65|
> |    |FLAIR+FARE    |79.21  |81.11  |-22.44 |62.81  |51.39  |-54.02 |42.76|
> |    |SAGE  |73.52  |69.56  |-33.16 |63.96  |54.31  |-40.20 |46.32|
> |    |SAGE+FARE |**80.07**  |**82.42**  |**-17.97** |**70.91**  |**66.75**  |**-30.97** |**60.59**|
> |S-CIFAR100 |FLAIR  |3.05   |1.00   |**-0.00**  |2.94   |1.00   |**-0.00**  |1.00|
> |    |FLAIR+FARE    |3.15   |1.00   |-1.18  |3.03   |1.00   |-1.12  |1.00|
> |    |SAGE  |63.20  |49.02  |-22.62 |**48.49**  |**35.59**  |-19.89 |28.98|
> |    |SAGE+FARE |**60.07**  |**44.65**  |-35.89 |46.00  |30.75  |-34.48 |**24.76**|
> |S-TinyImageNet |FLAIR  |1.46   |0.50   |**-0.00**   |1.46   |0.50   |**-0.00**   |0.50|
> |    |FLAIR+FARE    |1.46   |0.50   |**-0.00**   |1.46   |0.50   |**-0.00**   |0.50|
> |    |SAGE  |56.14  |44.72  |-13.54 |40.21  |31.95  |-9.49  |26.18|
> |    |SAGE+FARE |**61.03**  |**52.74**  |-23.52 |**43.59**  |**36.31**  |-21.98 |**30.51**|
>
> [1]Schlarmann C, Singh N D, Croce F, et al. Robust CLIP: unsupervised adversarial fine-tuning of vision embeddings for robust large vision-language models[C]//Proceedings of the 41st International Conference on Machine Learning. 2024: 43685-43704.

---

> ### Author Response · Authors · 2025-11-20
>
> **Q10: Exploring zero-shot adversarial robustness of SAGE.**
>
> **A10:** Following prior work, we trained on Tiny-ImageNet and evaluated zero-shot performance on Tiny-ImageNet, CIFAR-10, Food-101, DTD, EuroSAT, and PCAM, covering general classification, fine-grained recognition, texture, satellite imagery, and medical images. We compared SAGE against the original CLIP and the zero-shot adversarial robustness baselines mentioned in the paper, including TeCoA, FARE, PMG-AFT, and TGA-ZSR. The results in Tables S15 and S16 show that SAGE achieves competitive zero-shot clean accuracy and robustness compared to CLIP and most zero-shot adversarial robustness approaches, indicating that, although not specifically designed for this setting, it can still achieve satisfactory performance.
>
> Table S15: Zero-shot adversarial robust accuracy across 6 datasets by training on Tiny-ImageNet.
>
> | Method   | Tiny-ImageNet | CIFAR-10 | Food101 | DTD   | EuroSAT | PCAM  | Average |
> |----------|---------------|----------|---------|-------|---------|--------|---------|
> | CLIP     | 0.88          | 2.42     | 6.60    | 2.02  | 0.05    | 0.11   | 2.01    |
> | TeCoA    | 37.57         | 30.30    | 14.76   | 17.45 | 12.14   | 47.39  | 26.60   |
> | FARE     | 23.88         | 21.25    | 10.97   | 10.96 | 0.14    | 10.17  | 12.90   |
> | PMG-AFT  | 47.11         | **46.01**    | 19.58   | 15.05 | 12.54   | 48.64  | 31.49   |
> | TGA-ZSR  | 63.95         | 61.45    | **33.97**   | **22.08** | **14.27**   | 47.76  | **40.58**   |
> | SAGE (k=0.01) | 44.42  | 36.07    | 11.64   | 14.31 | 10.86   | **50.02**  | 27.89   |
> | SAGE (k=0.1)  | **50.34**  | 40.32    | 12.71   | 14.31 | 11.49   | 49.95  | 29.85   |
>
>
> Table S16: Zero-shot clean accuracy across 6 datasets by training on Tiny-ImageNet.
>
> | Method   | Tiny-ImageNet | CIFAR-10 | Food101 | DTD   | EuroSAT | PCAM  | Average |
> |----------|---------------|----------|---------|-------|---------|--------|---------|
> | CLIP     | 57.26         | **88.06**    | **83.89**   | **40.69** | **42.59**   | **52.09**  | **60.76**   |
> | TeCoA    | 63.97         | 66.14    | 35.11   | 25.53 | 17.13   | 50.01  | 42.98   |
> | FARE     | **77.54**         | 87.58    | 70.02   | 36.33 | 22.69   | 46.07  | 56.71   |
> | PMG-AFT  | 67.11         | 74.62    | 37.47   | 21.17 | 17.76   | 49.99  | 44.69   |
> | TGA-ZSR  | 75.72         | 86.46    | 57.59   | 29.06 | 24.24   | 49.58  | 53.78   |
> | SAGE (k=0.01) | 68.35 | 68.76    | 25.47   | 20.37 | 15.33   | 50.02  | 41.38   |
> | SAGE (k=0.1)  | 72.62 | 71.62    | 27.25   | 21.22 | 16.56   | 50.01  | 43.21   |
>
>
> **Q11: Use of textual prompts.**
>
> **A11:** In our comparison with the baselines, we consistently use the template ''This is a photo of {}.'' Moreover, we thoroughly examined the original implementations of all baselines, and whenever different templates were employed, we adopted those specified in the corresponding papers to ensure faithful results.

---

> > ### Comment · Reviewer_j5be · 2025-11-21
> >
> > I have carefully reviewed the concerns raised by other reviewers and the authors' responses. Based on my personal assessment, the authors' comprehensive response and detailed additional experiments have successfully resolved almost all of my concerns.
> >
> > Regarding Response A2, since you have demonstrated that replacing BCE with CE resolves the performance collapse of baselines like FLAIR in the CLIP setting, I strongly recommend updating Table 2 to report these optimized CE-based results from Table S7. As the shift from ResNet to CLIP necessitates adapting methods for optimal performance, presenting collapsed results (~1% accuracy) may mislead readers; given that SAGE still outperforms these improved baselines, this adjustment will ensure a sound comparison and further strengthen the credibility of your empirical evaluation.
> > Furthermore, I suggest including the detailed analysis regarding the incompatibility of BCE in the CLIP setting in the Appendix J.4.2 "THE TRAINING DETAILS OF FLAIR"
> >
> > Regarding Response A5, I was referring to a typographical inconsistency in the subscript font style of the loss notation within the equation, rather than questioning the validity, so please double-check the formatting.
> >
> > Regarding Response A8, my original question was specifically about the drop in zero-shot capability immediately after the first task, to assess early-stage forgetting. However, Table S13 appears to report the final or aggregated performance. Could you clarify the performance drop specifically after Task 1? Also, could you clarify the specific evaluation protocol (e.g., attack method, perturbation budget) used for Tables S15?
> >
> >
> > Before finalizing my review, I would like to open a brief discussion (Please note that the following points are intended solely for exchange to hear your insights; they do not constitute weaknesses affecting my score, nor do they require any further experiments.):
> >
> > - It is well-established that adversarial training induces fundamentally different feature representations (e.g., perceptually aligned) compared to standard training. I am interested in your insight regarding the internal mechanism of AT + CIL: Do you believe that current constraints (distillation, replay buffers, learning parameters) genuinely preserve the underlying robust feature manifolds of previous tasks? Or do they primarily mitigate the symptom of forgetting (i.e., restoring performance via replay/or constraint) while the actual robust representations might still be degrading or shifting?
> > - AT + CIL inherently creates an extreme data-scarce scenario for past tasks. Given that AT is known to exacerbate robust fairness issues in imbalanced settings (and such issues persist even in balanced settings), the combination of these factors creates layers of compounded complexity, making it difficult to interpret the underlying behavior. Do you believe that the AT + CIL problem is fundamentally solvable through algorithmic constraints like SAGE, considering these stacked challenges?
> > - In your conclusion, you mentioned future work on efficiency and scalability. I would like to hear more details on your vision, for instance, do you have insights regarding challenging scenarios like CIL on ImageNet or Few-Shot CIL?
> > - If the paper is accepted, do you plan to release a maintained open-source repository (e.g., GitHub)?

---

> > > ### Author Response · Authors · 2025-11-22
> > >
> > > We truly appreciate your time in reviewing our response and your encouraging feedback.
> > >
> > > **Regarding Response A2:**
> > > We fully agree that updating the results by replacing BCE with the optimized CE-based version across all relevant tables is more appropriate. We will also include a detailed analysis of the incompatibility of BCE in the CLIP setting in Appendix J.4.2. These updates will be incorporated in the next version.
> > >
> > > **Regarding Response A5:** We will carefully review all notations.
> > >
> > > **Regarding Response A8:** Sorry for the misunderstanding. We further report zero-shot capability immediately after training the first task to assess early-stage forgetting. During training, we use PGD-2 with a perturbation strength of 1/255 and step size 1/255 on CIFAR-10, following the main experimental setup. To validate zero-shot capability, we follow TeCoA and TGA-ZSR using PGD-100 with a perturbation strength of 1/255 and step size 1/255 on Tiny-ImageNet for both Tables S13 and S17. This evaluation protocol is also used in Table S15. As shown in Table S17, robustness does not improve on the first task. Methods designed for zero-shot adversarial robustness show a slight drop in clean accuracy, while methods designed for class-incremental learning experience an even larger decrease. This pattern is expected because the lack of information makes robustness gains impossible at this stage, and full fine-tuning further pulls the CIL model toward CIFAR-10, leading to a sharp decrease in clean accuracy on Tiny-ImageNet.
> > >
> > > Table S17: Zero-shot adversarial robustness and clean accuracy after the first task.
> > > |   |CLIP| TeCoA | FARE | PMG-AFT | TGA-ZSR || R-LwF | R-LwF-MC | R-EWC-on | R-SI | R-RAPF | R-SG | R-Proof || FLAIR | SAGE|
> > > |------|------|------|------|------|------|------|------|------|------|------|------|------|------|------|------|------|
> > > |Robust (%) |0.88|0.46|1.00|0.41|1.31||0.60|0.73|0.72|1.14|1.29|0.29|0.19||0.73|0.69|
> > > |Clean (%) |57.26|52.07|53.19|52.24|48.82||1.15|1.97|1.43|0.60|4.04|0.43|1.19||1.98|1.68|

---

> > > ### Author Response · Authors · 2025-11-22
> > >
> > > **Regarding Discussion 1:** We thank the reviewer for raising this insightful question regarding the internal mechanism of adversarial training combined with continual learning (AT + CIL). Indeed, adversarial training is known to produce perceptually aligned and robust feature representations, and how these representations evolve under continual learning constraints remains an important open question.
> > >
> > > The current mechanisms, such as distillation, replay buffers, and carefully tuned learning parameters, primarily serve to mitigate catastrophic forgetting and maintain task performance. While they help retain the functional outcomes of robust features (e.g., classification accuracy under adversarial perturbations), they may not fully guarantee that the underlying robust feature manifolds remain perfectly preserved. Some gradual drift or reorganization of feature representations could still occur, even when overall performance appears stable.
> > >
> > > Studying the internal dynamics of these robust representations under continual learning is indeed an interesting direction for future work. Techniques such as representation similarity analysis, probing feature alignment across tasks, or manifold visualization could help elucidate whether these mechanisms genuinely preserve robust features or mainly act as performance-level remedies.
> > >
> > > **Regarding Discussion 2:** Indeed, AT + CIL naturally places past tasks in an extremely data-scarce regime, and adversarial training is known to amplify representation imbalance and robust fairness issues, even when the original data distribution is balanced. These compounded factors make the underlying behavior of AT + CIL difficult to disentangle fully, and we agree that this complexity extends beyond what any single algorithmic constraint can fully resolve.
> > >
> > > Our view is that methods like SAGE offer partial but meaningful progress: By selectively protecting critical parameters and constraining important directions, it can slow the drift of features, mitigate performance degradation, and maintain usable robust features under limited samples. They help stabilize representations and reduce forgetting by imposing structure on how adversarial and clean features evolve across tasks. However, it cannot fully eliminate cumulative drift, nor can it guarantee complete preservation of global robust features under extreme data scarcity or complex class distributions.
> > >
> > > **Regarding Discussion 3:** Adversarial training improves robustness but is computationally expensive, especially on large datasets like ImageNet, making full adversarial training impractical. As discussed in Q9, a robust backbone can further enhance performance. At the same time, our analysis in response to reviewer BwZM's Q4 (Table S6) shows that the last layers contribute a large portion of robustness while being cheap to optimize. This suggests a strategy for large-scale CIL: use a robust backbone for strong feature representations and adapt only the last layers to reduce computational cost.
> > >
> > > Few-shot CIL faces a different challenge: insufficient samples to generate meaningful adversarial examples. Existing work on few-shot adversarial robustness, such as APT [1], uses a robust encoder obtained through TeCoA training with lightweight prompt tuning to avoid large-scale adversarial data. Extending this idea to Few-shot CIL, one can pair a robust backbone with low-capacity task adapters such as prompts, small MLP heads, or masked last-layer updates, maintaining robustness without explicit adversarial generation.
> > >
> > > Although the challenges differ, both scenarios emphasize efficiency and scalability. A robust backbone provides baseline adversarial resistance and minimizes model updates, offering a practical direction for future robust class-incremental learning.
> > >
> > > [1] Li L, Guan H, Qiu J, et al. One prompt word is enough to boost adversarial robustness for pre-trained vision-language models[C]//Proceedings of the IEEE/CVF Conference on Computer Vision and Pattern Recognition. 2024: 24408-24419.
> > >
> > > **Regarding Discussion 4:** We provided the code in the supplementary material and will release a maintained open-source repository on GitHub upon paper acceptance.

---

> > > > ### Comment · Reviewer_j5be · 2025-11-22
> > > >
> > > > I appreciate the authors' detailed response and the insightful discussion, which have resolved all my concerns.
> > > > Provided that the promised updates are incorporated in the final version, I believe this work makes a solid contribution and deserves to be presented as a poster. Therefore, I am happy to increase my score to "Accept".

---

> > > > > ### Author Response · Authors · 2025-11-22
> > > > >
> > > > > Thank you for taking the time to read our response and increasing your score! We are glad to hear that the response addressed your concerns.

---

### Official Review · Reviewer_ZojM · 2025-10-26

**Soundness:** 2
**Presentation:** 2
**Contribution:** 2
**Rating:** 6
**Confidence:** 3

**Summary:**

This paper introduces SAGE (Selective parameter optimization for Adversarial training with GEometric constraint), a novel framework for Robust Class-Incremental Learning (RCIL) based on CLIP. The authors formalize RCIL as a new problem setting that jointly tackles catastrophic forgetting and adversarial robustness. SAGE selectively updates important parameters identified by gradient–weight products while enforcing a geometric-constraint-based contrastive loss to preserve feature structure and mitigate forgetting. Extensive experiments on CIFAR-10/100, STL-10, and Tiny-ImageNet demonstrate that SAGE achieves superior robustness and lower forgetting compared with both classical and CLIP-based CIL baselines.

**Strengths:**

1. The paper formally defines the RCIL problem for the first time and provides theoretical insight into the intrinsic conflict between adversarial robustness and knowledge retention.

2. SAGE integrates two elegant ideas: (1) selective parameter optimization based on gradient–weight importance to control stability-plasticity trade-off, and (2) a geometry-constrained symmetric contrastive loss that preserves inter-task feature consistency.

3. The authors conduct comprehensive experiments across four datasets and multiple baselines (e.g., FARE, R-LwF, FLAIR). The results are consistently strong, with up to 15–20% improvement in PGD/AA robustness and substantially reduced backward transfer (BWT). Ablation studies clearly demonstrate the contribution of each module, showing a careful empirical validation.

4. The use of Taylor expansions to derive the trade-off between robustness and stability gives the method a solid theoretical grounding, adding interpretability and rigor often missing in CIL literature.

**Weaknesses:**

1. The framework relies on PGD-based adversarial training, which is computationally expensive and may not scale to large-scale CLIP variants or datasets such as ImageNet-1K. No efficiency comparison or training-time analysis is provided.

2. Experiments are all performed on standard small-scale benchmarks (CIFAR/STL/Tiny-ImageNet). It remains unclear whether SAGE generalizes to realistic continual learning settings with large-scale, noisy, or multimodal data.

3. There is no comparison with recent lightweight adversarial training or robustness transfer methods that reduce computational overhead (e.g., free/fast adversarial training). Including such baselines would enhance the claim of practicality.

**Questions:**

See Weaknesses

---

> ### Author Response · Authors · 2025-11-20
>
> **Q1: Discussion of computational overhead and Scalability to large CLIP variants.**
> * **Q1-1: Discussion of computational overhead.**
>
> * **A1-1:** We previously compared the robustness and training FLOPs of our method with representative benchmark methods on S-STL10 (originally presented in Appendix Figure 5). Here, we present the results in Table S1. "R-X" denotes the continual learning method X trained with adversarial examples. The results demonstrate that our method achieves the highest adversarial robustness while requiring the lowest training FLOPs among all baselines.
>
> Table S1: Comparison of Robustness and Training FLOPs across different methods on S-STL10.
> |	|TeCoA | FARE | PMG-AFT | TGA-ZSR || R-LwF | R-LwF-MC | R-EWC-on | R-SI | R-RAPF | R-SG | R-Proof || FLAIR | SAGE|
> |------|------|------|------|------|------|------|------|------|------|------|------|------|------|------|------|
> |PGD $A_{last}$ (%) | 30.14 |43.02|30.06|41.51||28.16|47.01|25.59|27.50|14.25|30.19|15.29||48.26|**54.31**|
> |Flops (10^15) |1.26|1.78|2.29|2.81||1.96|2.00|1.30|**1.24**|1.31|1.44|**1.24**||2.17|**1.24**|
>
> To provide a clearer view of computational efficiency, we additionally report the total training time for three settings: (1) standard CIL baselines trained on clean data only (LwF, LwF-MC, EwC-on, SI, RAPF, SG, and Proof), (2) adversarially trained baselines (TeCoA, FARE, PMG-AFT, TGA-ZSR, R-LwF, R-LwF-MC, R-EwC-on, R-SI, R-RAPF, R-SG, and R-Proof), and (3) the proposed RCIL (FLAIR and SAGE). The total time (in seconds) for the entire training and validation process on S-CIFAR10 is reported in Table S2.
>
> Table S2: Comparison of Robustness and Total Time across different methods on S-CIFAR10.
> |	| LwF | LwF-MC | EWC-on | SI | RAPF | SG |Proof||TeCoA | FARE | PMG-AFT | TGA-ZSR || R-LwF | R-LwF-MC | R-EWC-on | R-SI | R-RAPF | R-SG | R-Proof | |FLAIR | SAGE|
> |------|------|------|------|------|------|------|------|------|------|------|------|------|------|------|------|------|------|------|------|------|------|-----|-----|
> |PGD $A_{last}$ (%) |6.91|7.05|3.94|4.52|2.28|14.90|2.16||9.20|27.33|9.11|15.70||19.38|24.08|19.38|19.38|19.34|18.46|13.61||32.31|**47.85**|
> |Total Time (s) |907|856|1138|813|847|**551**|1208||2924|5311|5018|5764||2184|4385|5453|3704|3731|3604|2999||3717|3716|
>
> Standard CIL baselines achieve the fastest training but at the cost of limited adversarial robustness. In comparison, SAGE operates within a similar computational budget while consistently providing notably stronger robustness.
>
> * **Q1-2: Scalability to large CLIP variants.**
> * **A1-2:** To further examine whether our method scales to large-scale CLIP variants, we extend the evaluation to ViT-L/14. The final results in Table S4 show that on the larger and stronger ViT-L/14 most methods experience even more severe forgetting under adversarial conditions. This is likely because full parameter tuning makes it harder for the model to retain previous tasks. SAGE still maintains a clear and stable advantage by improving both clean accuracy and adversarial robustness. The overall trend shows that the robustness and scalability of SAGE continue to hold when applied to larger backbones.
>
> Table S4: Evaluation of several methods on ViT-L/14 without memory. We report Clean, PGD-10 accuracy (%), and BWT on S-CIFAR10 under an attack strength of 1/255.
>
> |Type   |Method |Clean $\overline{A}$ |Clean $A_{last}$ | Clean $BWT$ | PGD $\overline{A}$  | PGD $A_{last}$ | PGD $BWT$|
> |------|------|-------|------|------|------|------|------|
> |AT |TeCoA  |45.27  |19.81  |-98.99 |43.88  |19.56  |-95.40|
> |    |FARE  |50.63  |13.85  |**-50.53** |42.26  |12.87  |**-47.75**|
> |R-CIL  |R-LwF-MC   |44.46  |20.20  |-93.38 |41.04  |19.22  |-89.83|
> |    |R-EWC-on  |39.15  |17.92  |-84.36 |37.05  |16.93  |-82.14|
> |    |R-SI  |44.39  |19.52  |-96.80 |42.86  |18.99  |-48.83|
> |R-CIL-CLIP |R-Proof    |29.90  |10.00  |-61.24 |28.44  |10.00  |-58.26|
> |RCIL   |FLAIR  |43.71  |19.72  |-95.08 |41.69  |19.30  |-91.24|
> |RCIL4CLIP  |SAGE   |**61.93**  |**34.19**  |-77.55 |**56.59**  |**28.36**  |-78.86|

---

> ### Author Response · Authors · 2025-11-20
>
> **Q2: Experimental results on noisy dataset.**
>
> **A2:** To evaluate the performance of our method, SAGE, in realistic continual learning scenarios, we extend our experiments to ImageNet-A [1], as shown in Table S8. This dataset consists of natural images that are particularly challenging for standard ImageNet-trained models, with diverse visual conditions and limited samples, making continual learning more sensitive to feature drift. ImageNet-A is divided into 10 tasks with 20 classes per task. The results show that SAGE consistently delivers higher robustness and lower forgetting on ImageNet-A, remaining stable even under severe distribution shifts and limited samples, demonstrating its effectiveness in realistic continual learning settings. We have reviewed the manuscript, and the experiment is provided in **Appendix G**.
>
> Table S8: Evaluation of several methods on ViT-B/32 without memory. We report Clean, PGD-10, Auto. accuracy (%), and BWT on S-ImageNet-A under an attack strength of 1/255.
>
> | Type         | Method   |Clean $\overline{A}$ | Clean $A_{last}$ | Clean $BWT$  |PGD $\overline{A}$ | PGD $A_{last}$ | PGD $BWT$ | Auto. $A_{last}$|
> |--------------|-----------|---------|------------|--------|----------|------------|--------|----------|
> | AT           | TeCoA       | 12.16      | 2.40   | -24.68   |0.81       | 1.63   | -3.72    |0.07|
> |              | FARE        | 24.53      | 12.82  | -15.50   |2.13       | 1.53   | -0.96    |0.26 |
> |              | PMG-AFT      | 12.39      | 2.40   | -25.64   |0.89       | 1.63   | -3.98    | 0.07 |
> |              | TGA-ZSR      | 13.69      | 3.50   | -29.26   |2.54       | 2.42   | -8.18    | 0.86 |
> | R-CIL        | R-LwF         | 7.48       | 3.75   | -22.70   |3.53       | 2.21   | -11.60   | 1.51 |
> |              | R-LwF-MC     | 1.14       | 0.39   | **-0.00**     |1.14       | 0.39   | **-0.00**     | 0.79 |
> |              | R-EWC-on      | 7.72       | 3.27   | -23.06   |3.71       | 2.60   | -11.51   | 1.71 |
> |              | R-SI         | 7.54       | 2.98   | -22.11   |3.46       | 2.21   | -10.52   | 1.51 |
> | R-CIL-CLIP  | R-RAPF       | 2.22       | 1.92   | -8.38    | 1.89       | 1.63   | -6.53    | 1.05 |
> |              | R-SG        | 5.36       | 3.01   | -5.60    |2.45       | 1.51   | -1.89    | 0.92 |
> |              | R-Proof       | 5.13       | 1.63   | -12.78   | 2.82       | 1.54   | -8.12    | 0.59 |
> | RCIL         | FLAIR        | 1.14       | 0.39   | **-0.00**     |1.14       | 0.39   | **-0.00**     | 0.79 |
> | RCIL4CLIP    | SAGE        | **20.45**      | **13.81**  | -8.61    |**8.91**       | 6.69   | -3.38    | **4.94** |
>
> [1]Hendrycks D, Zhao K, Basart S, et al. Natural adversarial examples[C]//Proceedings of the IEEE/CVF conference on computer vision and pattern recognition. 2021: 15262-15271.

---

> ### Author Response · Authors · 2025-11-20
>
> **Q3: Comparison with free adversarial training (FreeAT).**
>
> **A3:** To strengthen the assessment of practicality, we additionally include Free Adversarial Training (FreeAT) [1] as a lightweight robustness baseline. FreeAT avoids generating adversarial examples in separate steps and instead updates the model and the adversarial perturbation within the same backward pass. For FreeAT, we set the number of epochs consistent with those used for the other methods to ensure a fair comparison, using $m = 2$ in our experiments. The results in Table S9 show that across all sequential tasks and both attacks, SAGE consistently and significantly outperforms FreeAT, achieving stronger robustness and much less forgetting, making it a more suitable choice for robust class-incremental learning. We have reviewed the manuscript, and the experiment is provided in **Appendix H**.
>
> Table S9: Comparison with free adversarial training (FreeAT) on ViT-B/32 without memory. We report average PGD-10, Auto. accuracy (%) and BWT on S-CIFAR10, S-STL10, S-CIFAR100 and S-TinyImageNet under an attack strength of 1/255.
>
> | Dataset         | Method  |Clean $\overline{A}$ | Clean $A_{last}$ | Clean $BWT$  |PGD $\overline{A}$ | PGD $A_{last}$ | PGD $BWT$ | Auto. $A_{last}$|
> |-----------------|--------|---------|--------------|-----------|-------|------------|----------|-----------|
> | S-CIFAR10 |      FreeAT | 44.85   | 19.78      | -98.23    | 41.90 | 19.15      | -90.81   | 18.03     |
> |   | SAGE   | **72.36**   | **63.67**    | **-34.12**    | **61.75** | **47.85**      | **-42.27**   | **41.60**     |
> | S-STL10         | FreeAT | 54.66   | 39.81      | -73.77    | 44.63 | 24.49      | -80.11   | 17.74     |
> || SAGE   | **73.52**   | **69.56**    | **-33.16**    | **63.96** | **54.31**      | **-40.20**   | **46.32**     |
> |  S-CIFAR100    | FreeAT | 26.68   | 9.38       | -89.61    | 20.85 | 7.78       | -69.96   | 7.36      |
> || SAGE   | **63.20**   | **49.02**    | **-22.62**    | **48.49** | **35.59**      | **-19.89**   | **28.98**     |
> |  S-TinyImageNet  | FreeAT | 21.36   | 7.65       | -70.81    | 12.71 | 4.37       | -43.34   | 4.23      |
> |   | SAGE   | **56.14**   | **44.72**     | **-13.54**    | **40.21** | **31.95**      | **-9.49**    | **26.18**     |
>
>
> [1]Shafahi A, Najibi M, Ghiasi A, et al. Adversarial training for free![C]//Proceedings of the 33rd International Conference on Neural Information Processing Systems. 2019: 3358-3369.

---

### Official Review · Reviewer_dSoa · 2025-10-31

**Soundness:** 2
**Presentation:** 4
**Contribution:** 3
**Rating:** 4
**Confidence:** 3

**Summary:**

This paper studies robust class-incremental learning (RCIL), which jointly tackles class-incremental learning and adversarial robustness. It formulates RCIL as optimizing two objectives—mitigating forgetting and improving robustness to input perturbations—and proposes a novel method named SAGE to address this problem. The method has two components: selective parameter optimization, which updates only high-importance parameters based on an importance score, and a geometric constraint that introduces a loss jointly optimizing the two potentially conflicting RCIL objectives. The effectiveness of SAGE in both class-incremental performance and adversarial robustness is demonstrated across four datasets.

**Strengths:**

- Formalizes the RCIL problem and explicitly states the trade-off between mitigating forgetting and enhancing adversarial robustness.
- The proposed method-consisting of a contrastive loss and selective parameter optimization-is easy to adopt without complex implementation.
- Specifically, the geometric-constraint guided contrastive loss is theoretically well-grounded and effectively addresses both robustness and forgetting in RCIL jointly. The ablation study on contrastive loss clearly demonstrates the effectiveness of the loss.

**Weaknesses:**

- **Questionable novelty of the problem formalization** The paper presents itself as the first to formalize Robust Class-Incremental Learning; however, the cited literature seems to indicate that closely related settings have been explored. So it remains somewhat uncertain whether the current formalization represents a clear contribution.
- **Observation section** The points presented in the observation section appear to reflect widely accepted views in CIL and adversarial robustness when considered separately, and presenting them together as Eq. (8) may not, on its own, constitute a distinct observation of the paper. In addition, the claim that the two terms in Eq. (8) are *contradictory* and involve a trade-off would need more detailed justification.
- **Experimental Setup** is the primary weakness of the paper.
  1. **Weak PGD configurations**  Training uses only PGD-2 and evaluation uses PGD-10, which are relatively mild settings and insufficient to establish robustness claims.
  2. **Too small attack strength**  Similarly, The main tables evaluate at an attack strength of $\epsilon = 1/255$, with only limited reporting at $\epsilon \in \{1/255, 2/255, 4/255\}$ (e.g., in Table 3). These budgets are much smaller than the $\epsilon = 8/255$ commonly used in related work on this problem, making it difficult to conclude that the method is adversarially robust.
  3. **Unexpectedly low baselines on S-CIFAR100/S-TinyImageNet**  Several methods show abnormally low accuracy and BWT on CIFAR100 and TinyImageNet. Notably, FLAIR is reported as 3.05% (clean) on CIFAR-100 with BWT = 0.00, despite the relativeness of CIFAR-10 and CIFAR-100. These anomalies raise concerns about the correctness of the baseline implementations or evaluation protocol for those settings.

**Questions:**

1. Could the authors clarify in what sense the two terms in Eq. (8) are contradictory? Or is this claim primarily supported by empirical evidence (e.g., Appendix B)?
2. Could the authors explain the choice of attack strength $\epsilon=1/255$? Reporting results at a stronger budget ($\epsilon=8/255$) would substantially strengthen the soundness of the paper.
3. Could the authors comment or explain the unexpectedly low baseline results?

---

> ### Author Response · Authors · 2025-11-20
>
> **Q1: The problem formalization.**
>
> **A1:** Previous methods [1,2] focus on ResNet-based architectures and lack the broader representation capabilities of vision-language models like CLIP. These prior RCIL studies did not consider the challenges of CLIP representations, such as the semantic similarity in CLIP's text embeddings. In contrast, we formalize Robust Class-Incremental Learning (RCIL) for class-incremental scenarios under adversarial attacks using CLIP, which introduces unique challenges due to the sequential learning of new classes and the distinct transfer and alignment properties of CLIP representations. Naive extensions of ResNet-based methods fail under this setting, as confirmed in our experiments.
>
> Our contribution is therefore twofold: (1) defining the new RCIL problem for sequential class learning under adversarial threats in a CLIP-based framework, and (2) proposing SAGE, which selectively optimizes parameters with geometric and contrastive constraints to preserve robustness and mitigate forgetting. These elements are essential for stable learning in this setting and are not addressed by prior ResNet-based CIL methods.
>
> [1]Bai T, Chen C, Lyu L, et al. Towards adversarially robust continual learning[C]//ICASSP 2023-2023 IEEE International Conference on Acoustics, Speech and Signal Processing (ICASSP). IEEE, 2023: 1-5.
>
> [2]Cho S, Lee H, Kim C. Enhancing Robustness in Incremental Learning with Adversarial Training[C]//Proceedings of the AAAI Conference on Artificial Intelligence. 2025, 39(3): 2518-2526.
>
>
> **Q2: The observation section.**
>
> **A2:** We thank the reviewer for the question. The main purpose of the observation section is not to present a single new equation, but to provide insight into the interaction between adversarial robustness and class-incremental learning. The claim that the two terms are 'contradictory' is primarily an insight from our analysis of their objectives, supported by empirical evidence. Adversarial training encourages feature representations to be locally invariant around each input, effectively 'flattening' the feature space for robustness. In contrast, class-incremental learning objectives aim to preserve class boundaries and maintain discriminative features from previous tasks. These objectives can pull feature representations in opposing directions, creating a trade-off that has not been explicitly discussed in prior work. We will also consider deleting or revising Eq. (8) to avoid confusion.
>
> This theoretical observation is further validated empirically in **Appendix B**, where naive combinations of adversarial training and standard CIL significantly increase forgetting or reduce robustness. Thus, the 'contradiction' reflects both the conflicting nature of the optimization objectives and the observed empirical behavior, motivating the need for methods like SAGE to balance these objectives effectively.
>
>
>
> **Q3: Experimental results of AutoAttack (Auto.).**
>
> **A3:** To comprehensively evaluate the model's adversarial robustness, we not only validate it under PGD attacks after training with PGD, but also conduct evaluations using AutoAttack. In all tables, we use 'AA' as an abbreviation for AutoAttack, which is explained in the experimental details. For clarity, we update the abbreviation 'AA' used in the tables to the more explicit 'Auto.'.
>
> **Q4: Experimental results under a stronger attack with perturbation strength 8/255.**
>
> **A4:** We follow prior work PMG-AFT [1], TGA-ZSR [2] by training at a perturbation strength of 1/255, and then evaluating at 1/255, 2/255, and 4/255. To further demonstrate the effectiveness of our method, we additionally evaluate at 8/255 on S-CIFAR10, as reported in Table S3. The results show that SAGE maintains the highest accuracy even under the strong 8/255 perturbation, demonstrating its superior adversarial robustness in continual learning settings.
>
> Table S3: Evaluation of several methods on ViT-B/32 without memory. We report PGD-10 accuracy (%) on S-CIFAR10 under an attack strength of 8/255.
> |	|TeCoA | FARE | PMG-AFT | TGA-ZSR || R-LwF | R-LwF-MC | R-EWC-on | R-SI | R-RAPF | R-SG | R-Proof || FLAIR | SAGE|
> |------|------|------|------|------|------|------|------|------|------|------|------|------|------|------|------|
> |PGD $A_{last}$ (%) | 0.04 |0.01|0.03|4.42||18.97|15.82|18.56|18.90|10.78|9.07|10.29||13.96|**19.03**|
> |PGD $\overline{A}$ |1.54|0.16|1.10|3.98||34.29|32.31|33.31|34.87|20.97|21.62|12.93||32.73|**35.03**|
>
> [1]Wang S, Zhang J, Yuan Z, et al. Pre-trained model guided fine-tuning for zero-shot adversarial robustness[C]//Proceedings of the IEEE/CVF conference on computer vision and pattern recognition. 2024: 24502-24511.
>
> [2]Yu L, Zhang H, Xu C. Text-guided attention is all you need for zero-shot robustness in vision-language models[J]. Advances in Neural Information Processing Systems, 2024, 37: 96424-96448.

---

> ### Author Response · Authors · 2025-11-20
>
> **Q5: Baseline performance analysis.**
>
> **A5:** To understand why FLAIR and R-LwF-MC behaved abnormally only in the CLIP-based continual learning setting, we analyzed the loss functions they rely on. Both FLAIR and R-LwF-MC adopt Binary Cross-Entropy (BCE) rather than Cross-Entropy (CE). This choice is effective for standard DNNs, but we speculate that it may be less suitable for CLIP, where text embeddings across classes tend to be highly correlated. The independence assumption of BCE may therefore be weakened, and the lack of normalization and inter-class competition could make it more challenging to separate semantically similar classes. Prior work[1] has also reported this limitation.
>
> To further verify that BCE is the source of the problem, we replaced the BCE loss in R-LwF-MC and FLAIR with CE loss while keeping all other implementations unchanged. The experimental results in Table S7 show that replacing BCE with CE eliminates the issues observed previously, leading to more stable and accurate predictions. We will provide additional experiments on this in the revised version.
>
> Table S7: Experiments on S-CIFAR100 replacing the BCE loss with CE loss in R-LwF-MC and FLAIR.
> |Method |Clean $\overline{A}$ |Clean $A_{last}$ |Clean $BWT$  |PGD $\overline{A}$   |PGD $A_{last}$   |PGD $BWT$    |Auto. $A_{last}$|
> |------|------|------|------|------|------|------|------|
> |R-LwF-MC   |28.24  |9.66   |-92.22 |24.82  |8.88   |-82.73 |8.85|
> |FLAIR  |28.57  |9.52   |-91.90 |24.88  |8.92   |-82.26 |8.91|
> |SAGE   |**63.20**  |**49.02**  |**-22.62** |**48.49**  |**35.59**  |**-19.89** |**28.98**|
>
> [1]Li Q, Xiao H, Shen L. BCE vs. CE in Deep Feature Learning[C]//Forty-second International Conference on Machine Learning.

---

### Official Review · Reviewer_z3NJ · 2025-10-31

**Soundness:** 2
**Presentation:** 3
**Contribution:** 2
**Rating:** 4
**Confidence:** 3

**Summary:**

The paper highlights a neglected issue in class-incremental learning (CIL): vulnerability to adversarial perturbations. It formalizes **Robust Class-Incremental Learning (RCIL)** and introduces **SAGE**—Selective parameter optimization for Adversarial training with GEometric constraint. SAGE selectively updates only critical parameters to preserve knowledge from previous tasks, offering parameter efficiency while resisting forgetting. A theoretically grounded geometric constraint, paired with a contrastive loss, preserves structural relationships among features, enabling stable, robust learning across increments under attack. Extensive experiments show SAGE improves adversarial robustness and simultaneously mitigates catastrophic forgetting, yielding more reliable, practical CIL models. Code is included in the supplementary material.

**Strengths:**

1. The writing is clear and easy to follow.
2. The experiments are fairly comprehensive and demonstrate the effectiveness of the proposed SAGE algorithm under the RCIL setting.

**Weaknesses:**

1. The importance and novelty of RCIL are not sufficiently justified. From my perspective, given prior CIL work, introducing RCIL feels natural—even trivial—and appears to be a simple combination of CIL with conventional adversarial robustness.
2. The baselines used to compare against SAGE are not described in adequate detail.
3. Much of the key notation is undefined—for example, $\mathcal{L} _{t}^{R}$ and $\mathcal{L} _{t}^{CIL}$ around lines 158–160.
4. Figure 1 conveys no information beyond what is already in its caption.
5. The attack strengths of 1/255, 2/255, and 4/255 are too small and are not standard perturbation radii in the adversarial robustness literature.

**Questions:**

See weaknesses.

---

> ### Author Response · Authors · 2025-11-20
>
> **Q1: The importance and novelty of RCIL.**
>
> **A1:** We respectfully disagree that RCIL is trivial. While inspired by CIL and adversarial robustness, it addresses unique challenges that arise from their combination. Adversarial attacks exacerbate catastrophic forgetting, cause representation drift across tasks, and destabilize learned features.  Naive application of conventional adversarial training to CIL not only fails to improve robustness but often worsens forgetting (as shown in Figure 4), demonstrating that the problem is non-trivial and requires new approaches.
>
> Our method, **SAGE**, is specifically designed to tackle these challenges. It employs **selective parameter optimization** to protect knowledge from previous tasks, a **geometric constraint** to preserve structural relationships among features, and a **contrastive loss** to stabilize representations under adversarial attacks. Ablation studies confirm that each component is essential for achieving robust and stable learning. RCIL, especially in a CLIP-based framework with sequentially arriving classes, is a **new and practically important problem** that prior ResNet-based methods cannot adequately address.
>
>
> **Q2: The experimental details of baselines.**
>
> **A2:** The experimental details of all baselines, including hyperparameters, loss functions, and textual prompts, are in **Appendix J**.
>
> **Q3: Several key notations.**
>
> **A3:** In Eq. (3), we describe the overall loss of RCIL, which integrates the objectives of adversarial robustness and CIL at task t. Specifically, $L_t^R$  and $L_t^{CIL}$ correspond to the adversarial robustness and CIL objectives, respectively. We will revise the manuscript to include clear definitions in the updated version.
>
>
> **Q4: Clarification of Figure 1.**
>
> **A4:** The main purpose of Figure 1 is to provide a visual overview of the RCIL framework and the interactions between adversarial robustness and CIL objectives. The figure offers a graphical illustration that can help readers quickly grasp the overall structure and workflow of the method.
>
> **Q5: Experimental results under a stronger attack with perturbation strength 8/255.**
>
> **A5:** We follow prior work PMG-AFT [1], TGA-ZSR [2] by training at a perturbation strength of 1/255, and then evaluating at 1/255, 2/255, and 4/255. To further demonstrate the effectiveness of our method, we additionally evaluate at 8/255 on S-CIFAR10, as reported in Table S3. The results show that SAGE maintains the highest accuracy even under the strong 8/255 perturbation, demonstrating its superior adversarial robustness in continual learning settings.
>
> Table S3: Evaluation of several methods on ViT-B/32 without memory. We report PGD-10 accuracy (%) on S-CIFAR10 under an attack strength of 8/255.
> |	|TeCoA | FARE | PMG-AFT | TGA-ZSR || R-LwF | R-LwF-MC | R-EWC-on | R-SI | R-RAPF | R-SG | R-Proof || FLAIR | SAGE|
> |------|------|------|------|------|------|------|------|------|------|------|------|------|------|------|------|
> |PGD $A_{last}$ (%) | 0.04 |0.01|0.03|4.42||18.97|15.82|18.56|18.90|10.78|9.07|10.29||13.96|**19.03**|
> |PGD $\overline{A}$ |1.54|0.16|1.10|3.98||34.29|32.31|33.31|34.87|20.97|21.62|12.93||32.73|**35.03**|
>
> [1]Wang S, Zhang J, Yuan Z, et al. Pre-trained model guided fine-tuning for zero-shot adversarial robustness[C]//Proceedings of the IEEE/CVF conference on computer vision and pattern recognition. 2024: 24502-24511.
>
> [2]Yu L, Zhang H, Xu C. Text-guided attention is all you need for zero-shot robustness in vision-language models[J]. Advances in Neural Information Processing Systems, 2024, 37: 96424-96448.

---

### Official Review · Reviewer_BwZM · 2025-11-01

**Soundness:** 3
**Presentation:** 3
**Contribution:** 3
**Rating:** 4
**Confidence:** 4

**Summary:**

The paper introduces a Robust Class-Incremental Learning (RCIL) method, requiring a CIL model to not only remember previous categories but also maintain adversarial robustness against malicious input perturbations across all classes learned so far. The proposed framework uses a CLIP-based approach to anchor the model in a naturally robust feature space, easing the conflict between learning new concepts and retaining robust knowledge of the old ones.

**Strengths:**

- Successfully formalizing and addressing the critical intersection of continuous learning (CIL) and AI security.

- Proposing a novel framework that systematically tackles the known challenge that Adversarial Training (AT) often leads to increased forgetting in CIL by leveraging the robust, fixed feature space of CLIP and employing a selective mask.

- Demonstrating extended sets of experiments and good empirical performance against the compared baselines.

**Weaknesses:**

-  Adversarial Training drastically increases training time compared to standard CIL baselines (like LwF or iCaRL) that use only clean data. This makes the method's practical application questionable for very large incremental tasks. Conduct a direct comparison of the total wall-clock training time required for the proposed RCIL method versus representative compared baselines (e.g., LwF, iCaRL, and the strongest integrated CIL+AT baseline) across the full sequence of tasks to understand the computational efficiency. Also, report the time complexity and discuss how the computational overhead introduced by the adversarial attacks and the selective mask scales with the number of classes and the number of incremental steps. or different attacks. This is critical for assessing the method's scalability.

- How much of the strong performance of the RCIL framework may be primarily due to the fixed, highly robust and generalizable features provided by the large, pre-trained CLIP encoder, rather than solely the selective parameter updates of the robust incremental learning method itself is unknown.



- Performance under black box attacks is unknown. Evaluate the final CIL model's adversarial robustness using the AutoAttack benchmark suite.

**Questions:**

- Conduct ablation experiments to study the mask. Study the evolution of mask across tasks. What specific layers or parameters are selected for adversarial robustness and clean accuracy. Does adversarial robustness primarily rely on protecting low-level features (early layers) while forgetting (clean performance) is more sensitive to high-level features (late layers)? This provides crucial insight into the mechanism of RCIL.

---

> ### Author Response · Authors · 2025-11-20
>
> **Q1: Discussion of computational overhead and total time.**
>
> **A1:** We previously compared the robustness and training FLOPs of our method with representative benchmark methods on S-STL10 (originally presented in Appendix Figure 5). Here, we present the results in Table S1. ''R-X'' denotes the continual learning method X trained with adversarial examples. The results demonstrate that our method achieves the highest adversarial robustness while requiring the lowest training FLOPs among all baselines.
>
> Table S1: Comparison of Robustness and Training FLOPs across different methods on S-STL10.
> |	|TeCoA | FARE | PMG-AFT | TGA-ZSR || R-LwF | R-LwF-MC | R-EWC-on | R-SI | R-RAPF | R-SG | R-Proof || FLAIR | SAGE|
> |------|------|------|------|------|------|------|------|------|------|------|------|------|------|------|------|
> |PGD $A_{last}$ (%) | 30.14 |43.02|30.06|41.51||28.16|47.01|25.59|27.50|14.25|30.19|15.29||48.26|**54.31**|
> |Flops (10^15) |1.26|1.78|2.29|2.81||1.96|2.00|1.30|**1.24**|1.31|1.44|**1.24**||2.17|**1.24**|
>
> To provide a clearer view of computational efficiency, we additionally report the total training time for three settings: (1) standard CIL baselines trained on clean data only (LwF, LwF-MC, EwC-on, SI, RAPF, SG, and Proof), (2) adversarially trained baselines (TeCoA, FARE, PMG-AFT, TGA-ZSR, R-LwF, R-LwF-MC, R-EwC-on, R-SI, R-RAPF, R-SG, and R-Proof), and (3) the proposed RCIL (FLAIR and SAGE). The total time (in seconds) for the entire training and validation process on S-CIFAR10 is reported in Table S2.
>
> Table S2: Comparison of Robustness and Total Time across different methods on S-CIFAR10.
> |	| LwF | LwF-MC | EWC-on | SI | RAPF | SG |Proof||TeCoA | FARE | PMG-AFT | TGA-ZSR || R-LwF | R-LwF-MC | R-EWC-on | R-SI | R-RAPF | R-SG | R-Proof | |FLAIR | SAGE|
> |------|------|------|------|------|------|------|------|------|------|------|------|------|------|------|------|------|------|------|------|------|------|-----|-----|
> |PGD $A_{last}$ (%) |6.91|7.05|3.94|4.52|2.28|14.90|2.16||9.20|27.33|9.11|15.70||19.38|24.08|19.38|19.38|19.34|18.46|13.61||32.31|**47.85**|
> |Total Time (s) |907|856|1138|813|847|**551**|1208||2924|5311|5018|5764||2184|4385|5453|3704|3731|3604|2999||3717|3716|
>
> Standard CIL baselines achieve the fastest training but at the cost of limited adversarial robustness. In comparison, SAGE operates within a similar computational budget while consistently providing notably stronger robustness.
>
> **Q2: Effect of CLIP features.**
>
> **A2:** Although CLIP processes strong zero-shot performance, previous studies on zero-shot adversarial robustness have shown that CLIP remains highly vulnerable to carefully crafted adversarial perturbations. For example, TeCoA [1] reports that under a PGD attack with a strength of 1/255, CLIP achieves only 9.57% adversarial robustness on CIFAR-10 and 4.55% on CIFAR-100. We also evaluate Continual-CLIP [2], a zero-shot, free-training approach, which exhibits similarly poor robustness. On S-CIFAR10, its PGD $A_{last}$ drops to just 2.62%, as shown in Table S5. These results suggest that CLIP itself does not provide highly robust features. Moreover, all baselines in our experiments use the same model architecture and leverage CLIP's generalizable features, yet SAGE still achieves superior performance, further validating the effectiveness of the proposed method.
>
> Table S5: Evaluation of Continual-CLIP on ViT-B/32. We report average PGD-10, Auto. accuracy (%) and BWT on S-CIFAR10, S-STL10, S-CIFAR100 and S-TinyImageNet under an attack strength of 1/255.
> | Dataset         | Method | PGD $\overline{A}$| PGD $A_{last}$| PGD $BWT$ | Auto. $A_{last}$ |
> |-----------------|--------|---------|--------------|-----------|-------|
> | S-CIFAR10       | Continual-CLIP   |8.19 | 2.62     | -4.09   | 0.00     |
> | S-STL10         |Continual-CLIP   |38.85	|30.29	|-9.13	|0.00		|
> | S-CIFAR100      | Continual-CLIP |2.00 |0.92 |-0.98 |0.00 |
> | S-TinyImageNet  | Continual-CLIP |0.80 |0.97 |-0.41 |0.00 |
>
> [1]Mao C, Geng S, Yang J, et al. Understanding Zero-shot Adversarial Robustness for Large-Scale Models[C]//The Eleventh International Conference on Learning Representations.
>
> [2]Thengane V, Khan S, Hayat M, et al. Clip model is an efficient continual learner[J]. arXiv preprint arXiv:2210.03114, 2022.
>
> **Q3: Experimental results of AutoAttack (Auto.).**
>
> **A3:** To comprehensively evaluate the model's adversarial robustness, we not only validate it under PGD attacks after training with PGD, but also conduct evaluations using AutoAttack. In all tables, we use 'AA' as an abbreviation for AutoAttack, which is explained in the experimental details. For clarity, we update the abbreviation 'AA' used in the tables to the more explicit 'Auto.'.

---

> ### Author Response · Authors · 2025-11-20
>
> **Q4: The mechanism of RCIL.**
>
> **A4:** To further analyze which layers contribute most to robustness in RCIL, we grouped the residual blocks of ViT-B/32 into early layers (blocks 0–3), middle layers (blocks 4–7), and last layers (blocks 8–11). We then conducted layer-wise ablation experiments by updating only the early, middle, or last layers, and evaluated the resulting model on clean and adversarial metrics, as summarized in Table S6.
>
> The results indicate that updating the last layers yields the largest gains in both adversarial robustness and clean accuracy. This is likely because updating deeper layers enables the model to learn more expressive and discriminative representations, which not only improves robustness but also helps mitigate forgetting in subsequent tasks, as reflected by a higher BWT. Updating only early or middle layers provides moderate improvements, while updating all layers achieves the best overall performance, highlighting the complementary contributions of different layer groups to both robustness and continual learning.
>
> Table S6: Layer-wise contribution analysis for robustness in RCIL.
>
> |Early layers   |Middle layers |Last layers |Clean $\overline{A}$ |Clean $A_{last}$ | Clean $BWT$ | PGD $\overline{A}$  | PGD $A_{last}$ | PGD $BWT$| AA $A_{last}$ |
> |------|------|------|------|------|-------|------|------|------|------|
> | √ |   |   |58.20  |32.09  |-72.81 |21.62  |8.14   |-37.43 |0.00   |
> |   | √ |   |59.59  |37.48  |-66.80 |39.87  |17.03  |-62.70 |0.51   |
> |   |   | √ |62.07  |41.12  |-50.34 |48.94  |26.18  |-48.41 |20.26  |
> | √ | √ | √ |72.36  |63.67  |-34.12 |61.75  |47.85  |-42.27 |41.60  |

---

### Author Response · Authors · 2025-11-20
**Total Summary-1**

**1、Summary:** We thank the reviewers for their positive and constructive comments. The reviewers agree that the proposed method is "novel", "easy to follow", and "theoretically well-grounded". They also agree that our work "effectively addresses both robustness and forgetting"", introduces a "well-structured and comprehensive framework", and can serve as "an important baseline and reference point" for future research on robust class-incremental learning (RCIL). In addition, they appreciate the "extended", "fairly comprehensive", and "logically organized and reproducible" experiments, noting that the performance is "consistently strong". The detailed feedback is as follows:

|Reviewers	|Aspects	|Comments|
|------|------|------|
|dSoa & j5be |Contribution|effectively addresses both robustness and forgetting; well-structured and comprehensive framework; a solid and credible foundation; an important baseline and reference point|
|BwZM & dSoa & ZojM & j5be|Method|novel; theoretically well-grounded; theoretical insight; simple yet effective; well-structured pseudo-code|
|BwZM & ZojM & j5be|Experiments|good empirical; careful empirical validation; solid; logically organized and reproducible; consistently strong|
|z3NJ |Clarity|easy to follow|


The reviewers' major comments suggest that additional analysis could provide a deeper understanding of the proposed method. Specifically, they recommended including: Discussion of computational overhead and total time. / Experimental results under a stronger attack with perturbation strength 8/255. / Experimental results on ViT-L/14. The additional analyses suggested by the reviewers are complementary and would enhance the understanding of our proposed method.

**2、More analysis of the proposed method:**

* 2.1 Additional computational overhead and total time of the models in Tables S1 and S2.

Table S1: Comparison of Robustness and Training FLOPs across different methods on S-STL10.

|	|TeCoA | FARE | PMG-AFT | TGA-ZSR || R-LwF | R-LwF-MC | R-EWC-on | R-SI | R-RAPF | R-SG | R-Proof || FLAIR | SAGE|
|------|------|------|------|------|------|------|------|------|------|------|------|------|------|------|------|
|PGD $A_{last}$ (%) | 30.14 |43.02|30.06|41.51||28.16|47.01|25.59|27.50|14.25|30.19|15.29||48.26|**54.31**|
|Flops (10^15) |1.26|1.78|2.29|2.81||1.96|2.00|1.30|**1.24**|1.31|1.44|**1.24**||2.17|**1.24**|

Table S2: Comparison of Robustness and Total Time across different methods on S-CIFAR10.

|	| LwF | LwF-MC | EWC-on | SI | RAPF | SG |Proof||TeCoA | FARE | PMG-AFT | TGA-ZSR || R-LwF | R-LwF-MC | R-EWC-on | R-SI | R-RAPF | R-SG | R-Proof | |FLAIR | SAGE|
|------|------|------|------|------|------|------|------|------|------|------|------|------|------|------|------|------|------|------|------|------|------|-----|-----|
|PGD $A_{last} (%)$ |6.91|7.05|3.94|4.52|2.28|14.90|2.16||9.20|27.33|9.11|15.70||19.38|24.08|19.38|19.38|19.34|18.46|13.61||32.31|**47.85**|
|Total Time (s) |907|856|1138|813|847|**551**|1208||2924|5311|5018|5764||2184|4385|5453|3704|3731|3604|2999||3717|3716|

* 2.2 Experimental results under a stronger attack with perturbation strength 8/255.

Table S3: Evaluation of several methods on ViT-B/32 without memory. We report PGD-10 accuracy (%) on S-CIFAR10 under an attack strength of 8/255.

|	|TeCoA | FARE | PMG-AFT | TGA-ZSR || R-LwF | R-LwF-MC | R-EWC-on | R-SI | R-RAPF | R-SG | R-Proof || FLAIR | SAGE|
|------|------|------|------|------|------|------|------|------|------|------|------|------|------|------|------|
|PGD $A_{last}$ (%) | 0.04 |0.01|0.03|4.42||18.97|15.82|18.56|18.90|10.78|9.07|10.29||13.96|**19.03**|
|PGD $\overline{A}$ |1.54|0.16|1.10|3.98||34.29|32.31|33.31|34.87|20.97|21.62|12.93||32.73|**35.03**|

* 2.3 Experimental results on ViT-L14.

Table S4: Evaluation of several methods on ViT-L/14 without memory. We report Clean, PGD-10 accuracy (%), and BWT on S-CIFAR10 under an attack strength of 1/255.

|Type   |Method |Clean $\overline{A}$ |Clean $A_{last}$ | Clean $BWT$ | PGD $\overline{A}$  | PGD $A_{last}$ | PGD $BWT$|
|------|------|-------|------|------|------|------|------|
|AT |TeCoA  |45.27  |19.81  |-98.99 |43.88  |19.56  |-95.40|
|    |FARE  |50.63  |13.85  |**-50.53** |42.26  |12.87  |**-47.75**|
|R-CIL  |R-LwF-MC   |44.46  |20.20  |-93.38 |41.04  |19.22  |-89.83|
|    |R-EWC-on  |39.15  |17.92  |-84.36 |37.05  |16.93  |-82.14|
|    |R-SI  |44.39  |19.52  |-96.80 |42.86  |18.99  |-48.83|
|R-CIL-CLIP |R-Proof    |29.90  |10.00  |-61.24 |28.44  |10.00  |-58.26|
|RCIL   |FLAIR  |43.71  |19.72  |-95.08 |41.69  |19.30  |-91.24|
|RCIL4CLIP  |SAGE   |**61.93**  |**34.19**  |-77.55 |**56.59**  |**28.36**  |-78.86|

---

> ### Author Response · Authors · 2025-11-20
> **Total Summary-2**
>
> **3. More explanation of the method.**
>
> * **The importance and novelty of RCIL.**  While inspired by CIL and adversarial robustness, it addresses unique challenges that arise from their combination. Adversarial attacks exacerbate catastrophic forgetting, cause representation drift across tasks, and destabilize learned features. Naive application of conventional adversarial training to CIL not only fails to improve robustness but often worsens forgetting (as shown in Figure 4), demonstrating that the problem is non-trivial and requires new approaches. Our method, SAGE, is specifically designed to tackle these challenges. It employs selective parameter optimization to protect knowledge from previous tasks, a geometric constraint to preserve structural relationships among features, and a contrastive loss to stabilize representations under adversarial attacks. Ablation studies confirm that each component is essential for achieving robust and stable learning. RCIL, especially in a CLIP-based framework with sequentially arriving classes, is a new and practically important problem that prior ResNet-based methods cannot adequately address.
>
> * **The purpose of the observation section.**  The main purpose of the observation section is not to present Eq. (8), but to provide insight into the interaction between adversarial robustness and class-incremental learning. The claim that the two terms of Eq. (8) are 'contradictory' is primarily an insight from our analysis of their objectives, supported by empirical evidence. Adversarial training encourages feature representations to be locally invariant around each input, effectively 'flattening' the feature space for robustness. In contrast, class-incremental learning objectives aim to preserve class boundaries and maintain discriminative features from previous tasks. These objectives can pull feature representations in opposing directions, creating a trade-off that has not been explicitly discussed in prior work. This theoretical observation is further validated empirically in **Appendix B**, where naive combinations of adversarial training and standard CIL significantly increase forgetting or reduce robustness. Thus, the 'contradiction' reflects both the conflicting nature of the optimization objectives and the observed empirical behavior, motivating the need for methods like SAGE to balance these objectives effectively.

---

### Meta-Review · Area_Chair_ti5a · 2026-01-03

**Summary:**

Several reviewers raised concerns about the novelty of the proposed RCIL4CLIP setting. Reviewers z3NJ and dSoa questioned whether RCIL should be viewed as a genuinely new problem, noting that it largely appears to be a straightforward combination of class-incremental learning and adversarial training. While the CLIP-based formulation introduces some practical differences, it was not fully convincing to all reviewers that this amounts to a fundamentally new setting. Moreover, most of the evaluations were done on rather small benchmarks, raising concerns whether the proposed method would be truly effective on larger-scale / longer tasks in which incremental learning becomes more important.

Relatedly, reviewers j5be and ZojM acknowledged the strong empirical results but noted that the proposed method can be seen as an extension or refinement of prior AT+CIL approaches, particularly FLAIR, adapted to the CLIP backbone. The authors’ rebuttal clearly explains why FLAIR does not transfer directly to CLIP (e.g., due to BCE loss incompatibility) and provides reasonable fixes. However, once these adaptations are made, the conceptual distinction between existing methods and SAGE appears relatively small.

Overall, while the paper is carefully implemented and supported by extensive experiments, the contribution feels incremental, driven more by CLIP-specific engineering choices than by a new conceptual insight. For this reason, I recommend rejection at this time, but encourage resubmission after clarifying the conceptual novelty and strengthening the distinction from prior AT+CIL methods such as FLAIR.

**Reviewer Concerns:**

[Addressed]
Concerns raised by BwZM and j5be regarding computational overhead and robustness evaluation were largely resolved through additional analyses. The authors provided detailed training time and FLOPs comparisons, evaluated stronger attacks (including larger perturbation budgets and AutoAttack), and clarified attack configurations. These additions significantly improved the transparency and completeness of the experimental evaluation.

The rebuttal also effectively addressed concerns about baseline validity, particularly those raised by j5be and dSoa. The authors identified the incompatibility of BCE loss with CLIP-based representations as the cause of the observed performance collapse for methods such as FLAIR and R-LwF-MC, and demonstrated that replacing BCE with CE restores reasonable performance. This explanation was convincing and resolved doubts about implementation correctness.

[Outstanding]
Questions raised by z3NJ and dSoa regarding the novelty of the RCIL formulation were only partially addressed. While the authors argue that the CLIP-based setting introduces new challenges, the rebuttal does not fully resolve the perception that RCIL4CLIP is a natural extension of existing AT+CIL frameworks rather than a fundamentally new problem.

Relatedly, despite the clarifications, the concern that SAGE represents an incremental refinement of prior methods (notably FLAIR) remains. As noted by j5be and ZojM, once prior methods are appropriately adapted to CLIP, the conceptual gap between these approaches and SAGE appears relatively narrow. This issue is not fully mitigated by additional experiments, as it primarily concerns conceptual novelty rather than empirical performance.

**Reviewer Scores:**

Reviewer BwZM
: Raised concerns primarily about computational overhead, scalability, and the contribution of CLIP features versus the proposed method. These concerns were addressed in the rebuttal through additional timing/FLOPs analysis and clarification that CLIP alone does not provide robustness. Given the thoroughness of the response, I believe this reviewer would likely have maintained their original score or increased it slightly, though their concerns about scalability may still limit a stronger endorsement.

Reviewer z3NJ
: Reviewer z3NJ was skeptical about the novelty and importance of RCIL, viewing it as a relatively natural combination of CIL and adversarial robustness. While the rebuttal clarified the motivation and provided additional experiments, the core concern about conceptual novelty was not fully resolved. As this concern is largely conceptual rather than experimental, I believe this reviewer would likely maintain their original score, remaining marginally below the acceptance threshold.

Reviewer dSoa
: Reviewer dSoa expressed mixed views, acknowledging the clarity and empirical strength of the method while questioning the novelty of the problem formulation and the robustness evaluation setup. The rebuttal addressed several technical points, including stronger attacks, AutoAttack results, and baseline anomalies, which likely alleviated some concerns. However, questions about the distinctiveness of the RCIL formulation may persist. I expect this reviewer would slightly increase their score, but still remain cautious.

Reviewer ZojM
: Reviewer ZojM viewed the paper more favorably overall, highlighting strong empirical results and solid theoretical grounding, while expressing concerns about efficiency and scalability. The rebuttal directly addressed these points with additional experiments and analyses. Given this alignment, I believe this reviewer would likely maintain or modestly increase their score, remaining on the acceptance-leaning side.

Reviewer j5be
: Reviewer j5be was initially positive but raised detailed technical concerns, particularly regarding baseline validity and experimental clarity. These issues were mostly addressed in the rebuttal, and the reviewer explicitly acknowledged that their concerns had been resolved and increased their score to accept. Therefore, I expect that the reviewer would have maintained the acceptance score.

---

### Decision · Program_Chairs · 2026-01-26

Reject